# BRCA1/BARD1 ubiquitinates PCNA in unperturbed conditions to promote continuous DNA synthesis

Daniel Salas-Lloret [1,7], Néstor García-Rodríguez [2,3,7], Emily Soto-Hidalgo [3], Lourdes González-Vinceiro[3], Carmen Espejo-Serrano[3], Lisanne Giebel[1], María Luisa Mateos-Martín [4], Arnoud H. de Ru[5], Peter A. van Veelen [5], Pablo Huertas [2,3], Alfred C. O. Vertegaal [1] & Román González-Prieto [1,3,6] ✉

Deficiencies in the BRCA1 tumor suppressor gene are the main cause of hereditary breast and ovarian cancer. BRCA1 is involved in the Homologous Recombination DNA repair pathway and, together with BARD1, forms a heterodimer with ubiquitin E3 activity. The relevance of the BRCA1/BARD1 ubiquitin E3 activity for tumor suppression and DNA repair remains controversial. Here, we observe that the BRCA1/BARD1 ubiquitin E3 activity is not required for Homologous Recombination or resistance to Olaparib. Using TULIP2 methodology, which enables the direct identification of E3-specific ubiquitination substrates, we identify substrates for BRCA1/BARD1. We find that PCNA is ubiquitinated by BRCA1/BARD1 in unperturbed conditions independently of RAD18. PCNA ubiquitination by BRCA1/BARD1 avoids the formation of ssDNA gaps during DNA replication and promotes continuous DNA synthesis. These results provide additional insight about the importance of BRCA1/BARD1 E3 activity in Homologous Recombination.

Breast cancer susceptibility type 1 (BRCA1) binds its partner BRCA1-Associated RING Domain 1 (BARD1) to form an obligated and multi-functional heterodimer with ubiquitin E3 ligase activity[1-3]. The BRCA1/BARD1 heterodimer acts as tumor suppressor and maintains genome stability, generally, by repairing deleterious double-strand DNA breaks (DSBs) via error-free homologous recombination (HR)[4,5]. DSBs can originate either from endogenous agents, such as reactive oxygens, replication fork progression and single-stranded DNA (ssDNA), or from environmental exposure to chemicals, ionizing radiation and ultraviolet light (UV)[6].

Germline mutations in BRCA1 and BARD1 are the main cause of hereditary breast and ovarian cancers, conferring a life-time probability of up to 90% for developing breast cancer and 50% for ovarian cancer[7-9]. Mice studies suggested that the E3 ligase activity is not essential to prevent tumor development[10], as the effects of patient-derived BRCA1 RING domain mutations such as C61G or C64G, which are required for tumor suppression and HR, are likely due to the disruption of the interaction of BRCA1 with BARD1[11]. Nevertheless, mutations in BARD1 that impair Histone 2A (H2A) ubiquitination have been identified in familial breast cancer[12].

BRCA1/BARD1 has a very well-established histone H2A ubiquitin ligase activity on lysines K127/K129[13-16]. This ubiquitination has been related to the maintenance of heterochromatin integrity, genetic stability, and senescence[2,13,16].

[1]Department of Cell and Chemical Biology, Leiden University Medical Centre, Leiden, The Netherlands. [2]Departamento de Genética, Facultad de Biología, Universidad de Sevilla, Sevilla, Spain. [3]Andalusian Centre for Regenerative Medicine and Molecular Biology (CABIMER), Universidad de Sevilla-CSIC-Universidad Pablo de Olavide-Junta de Andalucía, Sevilla, Spain. [4]Institute of Biomedicine of Seville, IBiS/Hospital Universitario Virgen del Rocío/CSIC/Universidad de Sevilla, Proteomics Facility, Sevilla, Spain. [5]Center for Proteomics and Metabolomics, Leiden University Medical Center, Leiden, The Netherlands. [6]Departamento de Biología Celular, Facultad de Biología, Universidad de Sevilla, Sevilla, Spain. [7]These authors contributed equally: Daniel Salas-Lloret, Néstor García-Rodríguez. ✉e-mail: roman.gonzalez@cabimer.es

Up to date, the relevance of the BRCA1/BARD1 E3 activity for DSBs repair, tumor suppression, and resistance to PARP inhibitors and platinum-based compounds is still controversial. Histone H2A K127/129 ubiquitination is required for DNA DSB repair by HR and RAD51 focus formation[2]. However, deficiencies in histone H2A ubiquitination by BRCA1/BARD1 can be rescued by other ubiquitin E3s such as RNF168[17,18]. While a study from the Morris group[2] using siRNA-based knockdowns in HeLa cells showed that the BRCA1/BARD1 E3 activity was required for resistance to agents such as the PARP inhibitor Olaparib and cisplatin, a degrON-based strategy on HCT116 cells showed that the BRCA1/BARD1 E3 activity was dispensable for resistance to Olaparib[19], consistent with most of the published literature[10,11,19,20].

Moreover, many research groups have addressed the challenge of identifying BRCA1/BARD1 ubiquitination substrates. However, all these studies have relied on indirect evidence to identify putative BRCA1/BARD1 ubiquitination substrates, by either overexpressing or depleting BRCA1 and identification of changes in the ubiquitin proteome by mass spectrometry-based proteomics[16,21–25]. However, since BRCA1/BARD1 plays an important role in several signaling pathways including cell cycle regulation, replication fork protection, gene transcription regulation, and DNA damage repair[5,26], these approaches could indirectly affect the ubiquitination state of proteins. Recently, we developed the TULIP and TULIP2 methodologies[27,28] which enable the unambiguous identification of direct ubiquitination substrates for an E3 of interest.

Here, we investigate the role of the BRCA1/BARD1 heterodimer E3 activity in the non-tumoral cell line RPE1 and apply the TULIP2 methodology to identify novel substrates for ubiquitination by BRCA1/BARD1. Combined, our data show that BRCA1/BARD1 promotes PCNA ubiquitination in unperturbed conditions to facilitate continuous DNA synthesis and provides insights about the ubiquitin E3 function of BRCA1/BARD1 during HR.

## Results

### BRCA1/BARD1 ubiquitin E3 activity is dispensable for DSB repair and resistance to treatment with Olaparib

The relevance of the BRCA1/BARD1 ubiquitin E3 activity for HR in terms of DNA DSB repair and treatment resistance remains contradictory in current literature, due to the use of different approaches involving diverse cancer cell lines and models[2,10,11,19,20]. To address this controversy, we employed RPE1 $TP53^{-/-}$ (Parental) and RPE1 $TP53^{-/-}$ $BRCA1^{-/-}$ (BRCA1-KO) cells[3] rescued or not with GFP-tagged BRCA1 wild type or I26A mutant at similar expression levels to the Parental cells BRCA1 levels (Supplementary Fig. 1a). The BRCA1 I26A mutant abrogates the BRCA1/BARD1 E3 activity without disrupting the formation of the BRCA1/BARD1 heterodimer (Supplementary Fig. 1b)[29,30]. We decided to test the proficiency in HR of the ubiquitin E3 activity BRCA1 mutant I26A in non-tumoral cells both in terms of resistance to Olaparib (Fig. 1a) and RAD51 focus formation in response to Ionizing Radiation (IR) (Fig. 1b, c). As expected, BRCA1-KO cells were sensitive to Olaparib and deficient in RAD51 focus formation in response to IR, and BRCA1-GFP rescued cells could restore both phenotypes to the Parental levels. Importantly, BRCA1-I26A-GFP rescued cells could also restore Olaparib resistance and RAD51 focus to the Parental levels, which is consistent with most of the published literature about the relevance of BRCA1 ubiquitin E3 activity in HR in DSB repair and resistance to Olaparib and Platinum-based compounds[10,11,17,19,20].

### BRCA1/BARD1 ubiquitinates PCNA in a constitutive manner

Previous studies aiming to identify BRCA1/BARD1 ubiquitination substrates are based on indirect evidence[14,16,31]. Moreover, the ubiquitination of histone H2A-K127/K129, which is the best studied BRCA1/BARD1 ubiquitination substrate[14,16], can be rescued by RNF168, another E3 enzyme, upon BRCA1 deficiencies[18]. Thus, we hypothesized that BRCA1/BARD1 ubiquitination substrates other than histone H2A isoforms exist.

We recently developed the TULIP2 methodology for the systematic application of Ubiquitin Activated Interaction Traps[27,28,32], which enables the identification of E3-specific targets in a direct manner. The rationale behind this approach is that if we have a linear fusion between an E3 and ubiquitin, this E3 could use its fused ubiquitin to modify a substrate protein. This strategy enables the co-purification and subsequent identification by mass spectrometry-based proteomics of the E3 together with ubiquitin and the covalently modified substrate. Therefore, we decided to apply TULIP2 methodology to identify BRCA1/BARD1-specific ubiquitination substrates. Three different TULIP2 constructs were generated to identify BRCA1 direct ubiquitination targets: BRCA1-WT-TULIP2, BRCA1-I26A-TULIP2, and BRCA1-WT-TULIP2ΔGG (Supplementary Fig. 1c; Fig. 1d). Both BRCA1-I26A and BRCA1-WT-TULIP2ΔGG served as negative controls. While BRCA1-I26A is a catalytic dead mutant without E3 activity, BRCA1-WT-TULIP2ΔGG fused ubiquitin cannot be conjugated to target proteins due to lack of the C-terminal diGly motif (Supplementary Fig. 1c).

First, we confirmed the functionality of the BRCA1-TULIP2 constructs for HR by introducing them in BRCA1-KO cells in a stable inducible manner and testing their resistance to Olaparib treatment (Supplementary Fig. 1e), and RAD51 focus formation in response to IR (Supplementary Fig. 1f, g). As previously seen for the GFP-tagged constructs, BRCA1-KO cells were hypersensitive to Olaparib and failed to form RAD51 focus in response to IR. However, BRCA1-KO cells stably expressing either BRCA1-WT-TULIP2 or BRCA1-I26A-TULIP2 completely rescued both Olaparib sensitivity and deficiency in RAD51 focus formation (Supplementary Fig. 1e–g). Besides, BRCA1-TULIP2 constructs co-localized with RAD51 similarly to endogenous BRCA1 (Supplementary Fig. 1f). These results corroborate the functionality of BRCA1-TULIP2 constructs and indicate a minor role of BRCA1/BARD1 E3 activity on HR pathway for DNA DSB repair and Olaparib resistance.

Next, we performed the TULIP2 assay to identify BRCA1 ubiquitin E3 activity-specific substrates using our BRCA1-TULIP2 cell lines (Supplementary Fig. 1c) and performed mass spectrometry-based proteomics analysis (Supplementary Data 1; Supplementary Fig. 1h-j). Analysis by immunoblotting failed to show a clear smear up from the wild type and I26A corresponding to BRCA1-TULIP2 conjugates as expected from functional TULIPs (Supplementary Fig. 1d)[27,28,33]. Moreover, proteomics analysis was unable to identify neither histone H2A nor macro-H2A as BRCA1-TULIP2 substrates, which would have served as positive controls, and the top hit was RAB43, which is not a nuclear protein. We concluded that the BRCA1-TULIP2 constructs were functional for BRCA1 activity but not regarding the TULIP2 assay, likely due to steric hindrance.

Nevertheless, a recent cryo-EM study has shown that it is BARD1, and not BRCA1, the heterodimer partner which positions the E2 enzyme and directs the ubiquitination of histone H2A[15]. Therefore, we made stable inducible BARD1-TULIP2 constructs and introduced them in Parental cells, including the wild type and ΔGG TULIP2 constructs. As an additional negative control, in this case, we introduced wild type BARD1-TULIP2 constructs in BRCA1-KO cells (Fig. 1d). Analysis by immunoblotting of the BARD1-TULIP2-expressing cells (Fig. 1e) revealed that: (1) The endogenous BARD1 levels were similar in Parental and BRCA1-KO cells; (2) the BARD1-TULIP2 construct levels were not detectable compared to endogenous BARD1 levels in the input samples, which is important to avoid overexpression artifacts; and (3) The BARD1-TULIP2 constructs were very efficiently enriched in the His-pulldown following TULIP2 methodology and a smear up from the wild type BARD1-TULIP2 construct corresponding to BARD1-TULIP2 conjugates, which was absent in the ΔGG control, could be detected. Mass spectrometry analysis of BARD1-TULIP2 samples (Supplementary Data 2; Fig. 1f–h) identified both histones H2A and macro-H2A as BARD1 ubiquitination substrates, serving as internal positive controls. Interestingly, we could identify another top hit, PCNA, as a

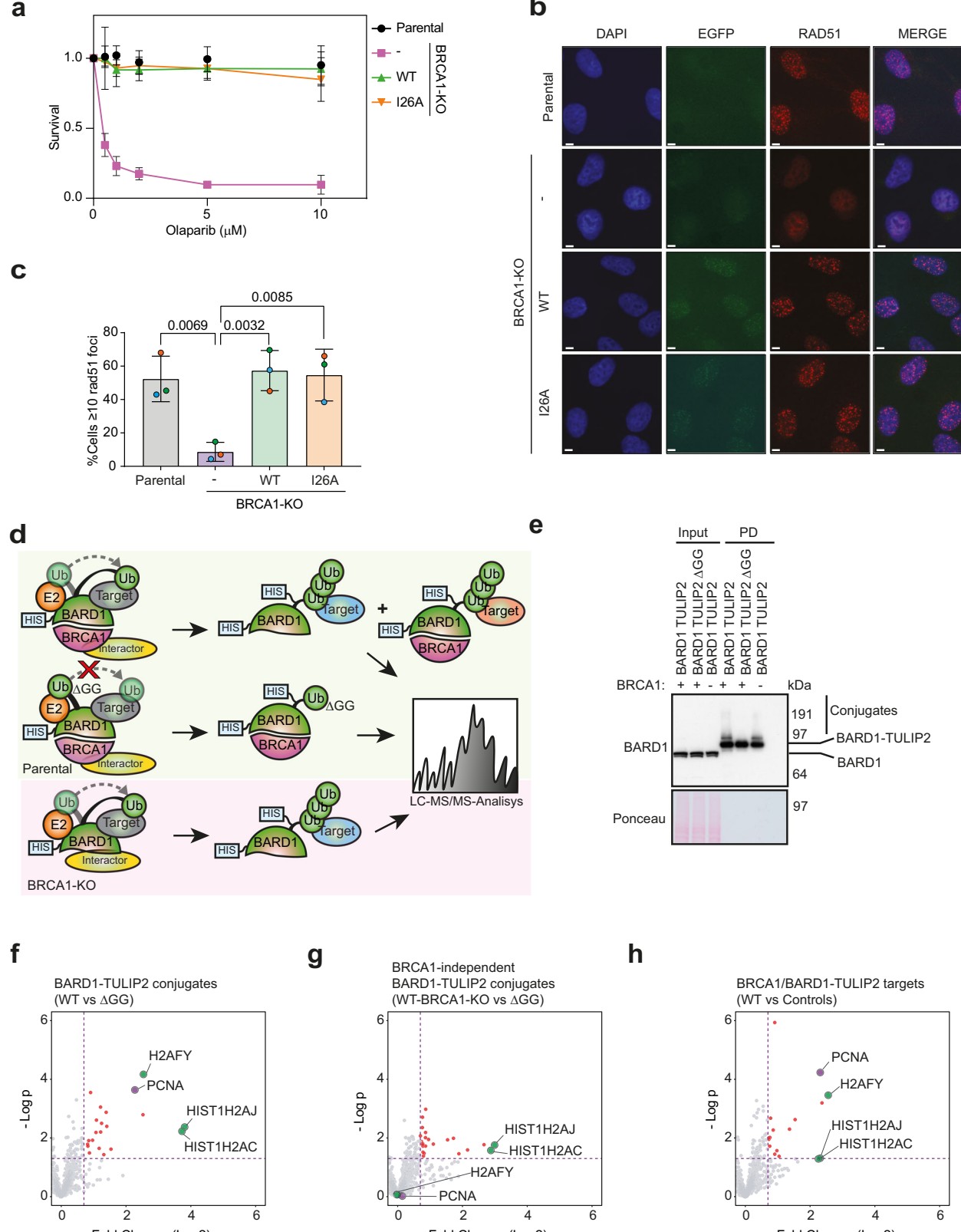

BRCA1/BARD1-specific ubiquitination substrate using TULIP2 methodology (Fig. 1h).

## BRCA1/BARD1 ubiquitinates PCNA to prevent ssDNA accumulation in unperturbed conditions

The proteomics spectra obtained in our BARD1-TULIP2 experiments did not enable the identification of the ubiquitin acceptor lysine in

PCNA for BRCA1/BARD1. K164 is the main ubiquitin acceptor site in PCNA, and this ubiquitination promotes the recruitment of Trans-Lesion Synthesis (TLS) DNA polymerases[34]. However, other acceptor lysines for ubiquitin have been identified in PCNA, which physiological relevance are unknown[35]. Thus, there was the possibility that the BRCA1/BARD1 heterodimer was ubiquitinating PCNA on a lysine other than K164. Therefore, we performed the BARD1-TULIP2 assay on RPE1

**Fig. 1 | BRCA1 E3 activity is not required for homologous recombination.**
**a** Clonogenic survival assay of Parental and BRCA1-KO cells complemented or not with either BRCA1-WT-GFP or BRCA1-I26A-GFP in response to different concentrations of Olaparib. Average and standard deviation of three independent experiments with three technical repeats are depicted ($N = 3$). **b, c** Immunofluorescence analysis Parental and BRCA1-KO cells complemented or nor with either BRCA1-WT-GFP or BRCA1-I26A-GFP after exposure to 10 Gy IR. Representative images (**b**) and quantification (**c**) of cells with RAD51 focus is shown. Size bars in fluorescence microscopy images represent 10 μm. Data corresponds to three independent experiments which value is

shown by an orange, green or blue circle respectively. Average and Standard Deviation is shown. Unpaired two-sided t-tests were performed with $p$ values shown in the figure (**d**) Cartoon depicting BARD1-TULIP2 rationale. **e** Analysis by immunoblotting of BARD1-TULIP2 samples. This analysis was performed for two out of four of the samples prior to LC-MS/MS analysis. **f–h**. Volcano plots depicting differences between each of the specified BARD1-TULIP2 constructs. $P$ value corresponds to unpaired two-sided t-tests. Histones H2A and macro-H2A and PCNA are labeled. Each dot represents a protein from Supplementary Data 2. Source data are provided as a Source Data file.

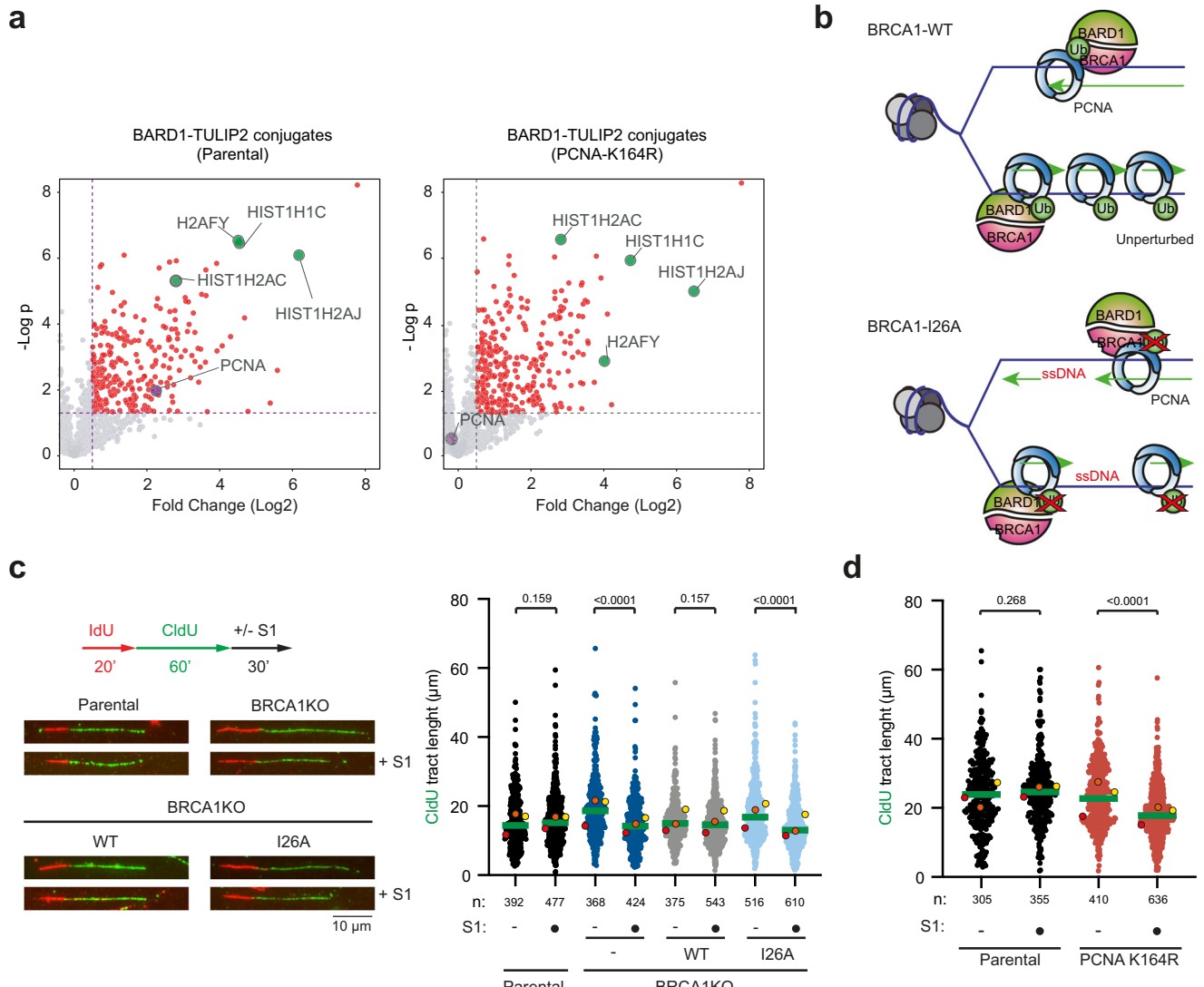

**Fig. 2 | BRCA1-BARD1 ubiquitinates PCNA on K164 for ssDNA gap suppression.**
**a** Volcano plots depicting BARD1-TULIP2 conjugates in the Parental and PCNA-K164R cell lines. PCNA is represented in purple and the Histones H2A and macro-H2A are in green. Each dot represents a protein from Supplementary Data 3. $P$ value corresponds to unpaired two-sided t-tests. **b** Cartoon depicting schematic representation of ssDNA accumulation in BRCA1-BARD1 E3 catalytic dead mutants. Formation of ssDNA gaps in unperturbed conditions in Parental, BRCA1-KO and BRCA1-WT and BRCA1-I26A rescued cells (**c**) and in Parental and PCNA-K164R

mutant cells (**d**). Top left: Scheme of the IdU/CldU pulse-labeling protocol, followed by S1 nuclease treatment. Bottom left: Representative images of DNA fibers. Graphs represent CldU tract lengths in the indicated cell lines with and without S1 nuclease treatment. Each dot represents one fiber and the green bar represents the median. n: indicates number of fibers measured from three biological independent experiments. $p$ values were calculated using the two-sided Mann-Whitney test. Colored dots indicate the median of each independent experiment. Source data are provided as a Source Data file.

$TP53^{-/-}$ $PCNA^{K164R/K164R}$ (PCNA-K164R) cells[36] (Supplementary Data 3; Fig. 2a). In this assay, we could identify histones H2A and macroH2A as BARD1 ubiquitination substrates. However, PCNA was lost as a BARD1-specific substrate, which indicates that BRCA1/BARD1 ubiquitinates PCNA on lysine K164.

PCNA K164 ubiquitination has recently been shown to participate in replication fork protection[36] and in preventing the accumulation of ssDNA gaps during DNA replication in unperturbed conditions[37], two functions that have been previously also attributed to BRCA1[38–41]. Different groups have shown that BRCA1-deficient cancer cells

accumulate ssDNA gaps and thus can be therapeutically exploited with PARP or REV1 inhibitors[39,42]. We hypothesized that BRCA1/BARD1 mediates PCNA ubiquitination to protect replication forks and avoid the accumulation of ssDNA gaps in unperturbed conditions. Therefore, mutations on either PCNA or BARD1-BRCA1 that compromise its E3 activity could lead to ssDNA accumulation and genetic instability (Fig. 2b).

To test BRCA1/BARD1 E3 activity on ssDNA gaps formation, we cultured different RPE1 cell lines with 10 µM CldU for 48 h and analyzed by immunofluorescence in non-denaturing conditions using an anti-BrdU antibody that will only recognize CldU present in ssDNA gaps[38] (Supplementary Fig. 2a). In line with previous research, we observed an increased number of ssDNA gaps in BRCA1-KO and PCNA K164R mutant RPE1 cells under normal growth conditions[36,39]. BRCA1 cells reconstituted with BRCA1-GFP suppressed the formation of ssDNA gaps to the Parental levels. However, complementation with the BRCA1-I26A-GFP mutant construct could not suppress ssDNA gap formation and showed similar levels of ssDNA foci to BRCA1-KO cells. However, this assay cannot distinguish replication-induced ssDNA gaps from ssDNA originating from other sources, such as DSB resection or other DNA repair intermediates. Alternatively, to corroborate that the E3 activity of the BRCA1/BARD1 heterodimer prevents the formation of ssDNA gaps, we performed the S1-nuclease assay on DNA fibers (Fig. 2c)[38,42]. In this assay, DNA fibers containing ssDNA gaps would be shortened after treatment with S1 nuclease, as it is the case in BRCA1-KO cells. Importantly, DNA fiber shortening was suppressed after complementation with wild-type BRCA1-GFP but not by BRCA1-I26A-GFP. Consistently, PCNA-K164R mutant cells also accumulated gaps behind the fork (Fig. 2d).

Subsequently, we performed the S1-nuclease assay on DNA fibers after mild replication stress (0.5 mM hydroxyurea (HU) for 2 h) (Supplementary Fig. 2b). Mild replication stress exacerbated the accumulation of ssDNA gaps in both BRCA1-KO and PCNA-K164R cells. As previously observed in unchallenged conditions (Fig. 2c, d), wild-type BRCA1-GFP, but not by BRCA1-I26A-GFP, could suppress the accumulation of ssDNA gaps, although the accumulation of ssDNA gaps was ameliorated in BRCA1-I26A-GFP cells compared to BRCA1-KO cells. Additionally, the increased replication fork speed observed in BRCA1-KO and BRCA1-I26A cells in response to mild HU treatment was also evident in PCNA-K164R cells when compared to Parental cells (Supplementary Fig. 2b, c).

## ssDNA gap-suppression function of PCNA Ubiquitination and BRCA1/BARD1 are in the same pathway

Next, we decided to investigate if the gap suppression functions of PCNA ubiquitination and BRCA1/BARD1 operate in the same pathway or through independent mechanisms. Thus, we performed the S1 nuclease fiber assay in unperturbed conditions in Parental or PCNA-K164R cells after treatment with siRNA control or siBRCA1 (Fig. 3a; Supplementary Fig. 3a). Both BRCA1-depleted Parental cells[38] and PCNA-K164R mutant cells showed increased replication fork speed and ssDNA gap accumulation in unperturbed conditions. However, BRCA1 depletion in PCNA-K164R cells did not cause an additional replication fork speed increase nor an additional accumulation of ssDNA gaps, thus indicating that the PCNA ubiquitination and BRCA1/BARD1 act in the same pathway regarding ssDNA gap suppression. In contrast, inhibition of PolΘ in PCNA-K164R cells with ART558 resulted in an additional formation of ssDNA gaps (Fig. 3b), which provide further evidence that non-epistatic interactions can be detected with this assay as previously observed for BRCA1-KO cells[43].

As lack of BRCA1 E3 activity causes the accumulation of replication associated ssDNA gaps, BRCA1-I26A mutant should be sensitive to replicative stress. Then, we decided to test the sensitivity to HU of the Parental and BRCA1-KO cells complemented or not with GFP-tagged constructs of either wild type or I26A mutant BRCA1. PCNA-K164R

mutant was also included (Fig. 3c). In contrast to what we observed for Olaparib (Fig. 1a), where sensitivity of BRCA1-deficient cells could be rescued by complementation with the BRCA1-I26A-GFP mutant, BRCA-I26A-GFP complemented cells were more sensitive to HU treatment than their wild-type counterpart. Notably, PCNA-K164R mutant cells were more sensitive than the I26A mutant, mirroring the sensitivity observed in BRCA1-deficient cells, indicative of the relevance of BRCA1 and PCNA ubiquitination in additional DNA damage repair or tolerance pathways. Accordingly, treating PCNA-K164R cells with an shRNA targeting BRCA1 caused additional sensitivity to replication stress (Fig. 3d, Supplementary Fig. 3b), at least at 2 mM HU, as previously described[36]. In contrast when introducing shRNA-resistant BRCA1-WT-GFP or BRCA1-I26A-GFP constructs in the PCNA-K164R cells (Fig. 3e), no additional sensitivity of the BRCA1 shRNA treatment was observed. Which indicates that the additional sensitivity to HU of BRCA1 knockdown in PCNA-K164R cells is not related to BRCA1 E3 activity.

We found intriguing that BRCA1-I26A cells were resistant to Olaparib (Fig. 1a), which induces ssDNA gaps[38,39], but not to gaps induced by HU (Fig. 3c, Supplementary Fig. 2c). Therefore, we decided to investigate how these treatments affected PCNA ubiquitination levels (Supplementary Figs. 3c, d). Analysis by immunoblotting of PCNA ubiquitination levels after treatment with Olaparib (Supplementary Fig. 3c), showed no effect. However, treatment with HR (Supplementary Fig. 3d) caused an accumulation of PCNA ubiquitination in Parental and BRCA1-WT-GFP rescued BRCA1-KO cells. This response to HU did not occur in BRCA1-I26A-GFP rescued cells. Interestingly, BRCA1-KO cells were able to respond to hydroxyurea treatment with an increase in relative PCNA ubiquitination levels, similar to the parental cells, which indicates that sensitivity of BRCA1-KO cells to HU is not related to PCNA ubiquitination.

Noteworthy, in unperturbed conditions (Supplementary Fig. 3e) PCNA-Ub level was reduced 60% in BRCA1-KO cells, and this decrease was partially rescued by BRCA1-WT. However, PCNA-Ub level was equally rescued by the BRCA1-I26A mutant, suggesting that the contribution of BRCA1 E3 activity to the total levels of ubiquitinated PCNA is negligible.

## BRCA1/BARD1 and RAD18 ubiquitinate PCNA-K164 in distinct scenarios

BRCA1 has been shown to promote the recruitment of RAD18 to chromatin in response to replication-blocking lesions[44]. Since RAD18 is considered the canonical ubiquitin ligase for PCNA in response to replication barriers[45–47] and protein-protein interactions between BRCA1 and RAD18 have been described[44], there was the possibility that BRCA1 absence was negatively affecting the E3 activity of RAD18 towards PCNA. To test this hypothesis, first, we performed iPOND experiments[48] to investigate if the absence of BRCA1 or its E3 activity was negatively affecting the recruitment of BRCA1 or RAD18 to replication forks (Fig. 4a, b; Supplementary Data 4). Neither the recruitment of RAD18 to replication forks was affected by the absence of BRCA1 nor the recruitment of BRCA1 was affected by the lack of E3 activity. Noteworthy, MDC1 was enriched in BRCA1-KO cells compared to Parental cells, indicative of DNA damage at the forks. Next, we irradiated with UV light Parental and BRCA1-KO cells and followed the accumulation of ubiquitinated PCNA in a time course manner (Supplementary Fig. 4a). No substantial differences in ubiquitinated PCNA accumulation kinetics could be observed after UV irradiation between Parental and BRCA1-KO cells. However, depleting the cells from RAD18, completely suppressed the accumulation of ubiquitinated PCNA after UV irradiation (Supplementary Fig. 4b), indicating that the absence of BRCA1 was not negatively affecting the E3 activity of RAD18 towards PCNA. Nevertheless, we decided to monitor the ubiquitin E3 activity of RAD18 towards PCNA in a direct manner.

Therefore, we made RAD18-TULIP2 constructs, introduced them in Parental and BRCA1-KO cells and performed the TULIP2 assay with

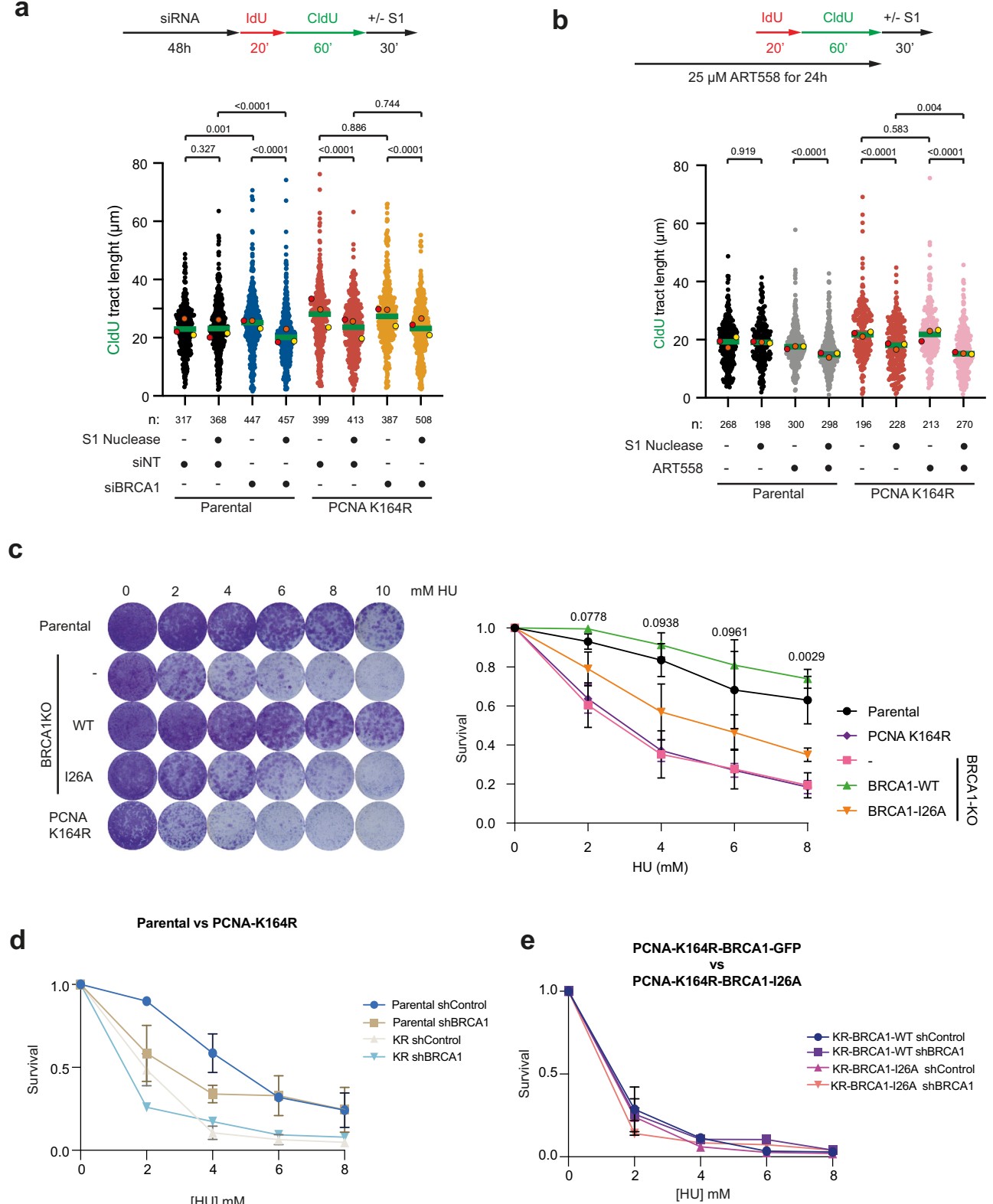

and without UV light irradiation (Supplementary Data 5; Fig. 4c). As expected, immunoblotting analysis (Supplementary Fig. 4c) showed that the wild type RAD18-TULIP2 samples formed a smear up from the RAD18-TULIP2 construct, indicative of its functionality, that was suppressed in the ΔGG negative controls. Mass spectrometry-based proteomics analysis of the non-irradiated samples (Fig. 4d) revealed that among the identified RAD18 substrates, PCNA was not present. This favors the hypothesis that BRCA1/BARD1 is, compared to RAD18, a

more active ubiquitin E3 enzyme for PCNA under unperturbed conditions (Fig. 1f). However, there was still the possibility that the absence of PCNA among the RAD18-specific substrates, regardless of the formation of the smear, was due to steric hindrance, similarly as previously observed for BRCA1 (Supplementary Fig. 1).

To serve as positive control, we irradiated with UV light RAD18-TULIP2 expressing Parental cells, performed the TULIP2 methodology and identified the substrates after UV irradiation (Supplementary

**Fig. 3 | PCNA-K164 ubiquitination and BRCA1 ssDNA gap suppression activities are in the same pathway.** Epistasis analysis of ssDNA gap formation. Cells were transfected with siRNA control (siNT) or siBRCA1 48 h before IdU and CldU labeling (**a**), or 24 h with the PolΘ inhibitor ART558 (**b**) followed by S1 nuclease treatment. Graphs represent CldU tract lengths in the indicated cell lines with and without S1 nuclease treatment. Each dot represents one fiber and the green bar represents the median. n: indicates number of fibers measured from three biological independent experiments. *p* values were calculated using two-sided Mann-Whitney test. Colored dots indicate the median of each independent experiment. **c** Survival assay of Parental, PCNA-K164R mutant and BRCA1-KO cell lines rescued with either BRCA1-

WT or BRCA1-I26A against different HU concentrations. Three independent experiments with two technical repeats were performed ($N = 3$). *P* values of unpaired two-sided t-tests between BRCA1-WT and BRCA1-I26A cells is indicated. **d** Survival assay after treatment with HU of Parental and PCNA-K164R mutant cells treated either with a control shRNA of BRCA1-targetting shRNA. **e** Survival assay after treatment with HU of PCNA-K164R mutant cells ectopically expressing an shRNA-resistant construct of BRCA1-WT or BRCA1-I26A treated either with a control shRNA of BRCA1-targetting shRNA. **c**–**e** Average and Standard Deviation of 3 independent experiments is depicted. Source data are provided as a Source Data file.

Data 5; Fig. 4e). PCNA became now a substrate while it was not in unperturbed conditions, confirming the functionality of the RAD18-TULIP2 construct. PCNA ubiquitination levels were lower in BRCA1 knockout cells compared to Parental cells in unperturbed conditions (Supplementary Fig. 3e). However, after UV irradiation, ubiquitination kinetics of PCNA were not affected by BRCA1 knockout (Supplementary Fig. 4a). Additionally, we confirmed by RAD18 knockdown that PCNA ubiquitination after UV exposure was RAD18-dependent and BRCA1-independent (Supplementary Fig. 4b). Finally, we performed the RAD18-TULIP2 assay in BRCA1-KO cells in unperturbed conditions and after UV irradiation (Fig. 4f, g). Mass spectrometry analysis revealed a substantial enrichment of PCNA as a RAD18-specific substrate compared to Parental cells in both unperturbed and UV conditions. The conjugation of PCNA to the RAD18-TULIP2 construct could be confirmed by immunoblotting (Fig. 4h), since PCNA antibody staining produced a band between 100 and 130 kDa, corresponding the size of a branched polypeptide consisting of RAD18-TULIP2 covalently bound to PCNA.

Next, we investigated how PCNA ubiquitination levels were affected by the absence of RAD18 and how these levels responded to HU treatment (Supplementary Fig. 4d). RAD18-KO cells had a big reduction on PCNA-Ub levels in unperturbed conditions, yet they were still able to respond to HU treatment. Importantly, the remaining PCNA ubiquitination levels in RAD18-KO cells sufficed to suppress the accumulation of ssDNA gaps in unperturbed conditions (Fig. 4i).

Taken together, our data indicate that BRCA1/BARD1, regarding the ssDNA gap-suppression function of PCNA ubiquitination, is the main E3 for PCNA in unperturbed conditions. However, it is likely that other E3s ubiquitinate PCNA in different scenarios also present in unperturbed conditions which are responsible of most of the ubiquitinated PCNA fraction. In the absence of BRCA1, RAD18 might compensate for PCNA ubiquitination levels in a scenario downstream the ssDNA gap formation.

Collectively, these data suggest that BRCA1/BARD1 and RAD18 share PCNA as substrate but act in different scenarios. While RAD18 would ubiquitinate PCNA when the replication fork encounters an obstacle on the DNA template, BRCA1/BARD1 would perform PCNA ubiquitination at low levels to facilitate continuous DNA synthesis during unperturbed conditions or upon induction of mild replicative stress that do not imply replication fork stalling, such as after mild HU treatment (Fig. 4j).

## Discussion

Here, we provide novel insight in the role of the BRCA1/BARD1 ubiquitin E3 activity in promoting continuous DNA synthesis during DNA replication. Our results (Fig. 1a–c) are in accordance with most studies and show that the E3 activity of BRCA1/BARD1 is not relevant for HR and Olaparib resistance[10,19,20]. However, we noticed that cells deficient in the BRCA1/BARD1 ubiquitin E3 activity were sensitive to HU (Fig. 3c) and accumulated more ssDNA gaps behind the replication forks (Fig. 2c, d; Supplementary Fig. 2), which indicates that deficiencies in the E3 activity of BRCA1/BARD1 do not affect the DSB repair function of HR but do affect the DNA replication function of HR[49].

Using TULIP2 methodology in unperturbed conditions, we identified PCNA-K164 as a ubiquitination substrate for BRCA1/BARD1 but not for RAD18 (Fig. 1d–h; Fig. 2a; Fig. 4b). Interestingly, previous studies show synthetic lethality in BRCA1 deficient cells with either RAD18 loss or the use of REV1 inhibitors[42,50]. PCNA is known to be mono-ubiquitinated at low levels in mammalian cells in unperturbed conditions[36,51,52], and PCNA ubiquitination levels in unperturbed conditions were reduced in BRCA1-KO cells (Supplementary Fig. 3e), although still some PCNA ubiquitination could be detected. Ectopic BRCA1-WT expression could partially rescue basal PCNA ubiquitination levels. However, BRCA1-I26A could also do it at the same extent, which suggests that the reduction of basal PCNA ubiquitination levels is due to replication fork collapse in the absence of BRCA1[53,54], as reflected in the accumulation of MDC1 at replication forks in BRCA1-KO cells compared to Parental cells (Fig. 4a; Supplementary Data 4). RAD18-KO also caused a big reduction in basal PCNA Ubiquitination levels (Supplementary Fig. 4d). Absence of RAD18 also causes transcription-dependent fork collapse[55]. Thus, we cannot exclude the possibility that the decrease in PCNA ubiquitination in RAD18-KO cells is also due to replication fork collapse. Moreover, BRCA1-KO also promotes the accumulation of R-loops[56], which can explain why activity of RAD18 towards PCNA increases by several orders of magnitude (Fig. 4f–g) but this ubiquitination is still unable to suppress the formation of replication-dependent ssDNA gaps. BRCA1 and RAD18 would be working in different scenarios arising in response to endogenous stresses but the absence of one would be promoting the accumulation of the toxic structures for which the other E3 is required. Thus, determining which fraction of the basal PCNA ubiquitination levels correspond to which E3 by quantifying this modification after the knockout of different E3s is misleading as can be due to an indirect effect (i.e., fork collapses occurring upon knockout of these E3s or re-wiring of the ubiquitination machinery).

Noteworthy, our TULIP2 approach (Figs. 1d–h, 2a, 3c–h) enabled us to access these hurdles by studying the activity of an E3 towards its substrates in a scenario where all the other E3s are present and with expression below the endogenous levels, avoiding the appearance of overexpression artifacts. With this approach, our data indicates that in unperturbed conditions BRCA1/BARD1 is the predominant E3 for PCNA, compared to RAD18, responsible of the replication-associated ssDNA gap suppression function of PCNA ubiquitination. Nevertheless, it is likely that other E3s, yet to be characterized, also modify PCNA in other situations arising in unperturbed conditions, given the big proportion of E3s in the human proteome which physiological relevance has not been studied yet.

E3s for PCNA other than RAD18 and BRCA1 have been described, including the Cullin 4 RING ligase, CRL4[57], or RNF214[58]. However, while CRL4 works synergistically with RAD18 and CRL4 knockdowns affect the accumulation of mono-ubiquitinated PCNA in response to UV irradiation[57], no differences were observed in PCNA ubiquitination in response to UV light in BRCA1-deficient cells (Supplementary Fig. 4a, b). This also supports the model that BRCA1/BARD1 and RAD18 modify PCNA in alternative pathways (Fig. 4g), without excluding the possibility that a crosstalk between BRCA1 and RAD18 exists as previously proposed[44].

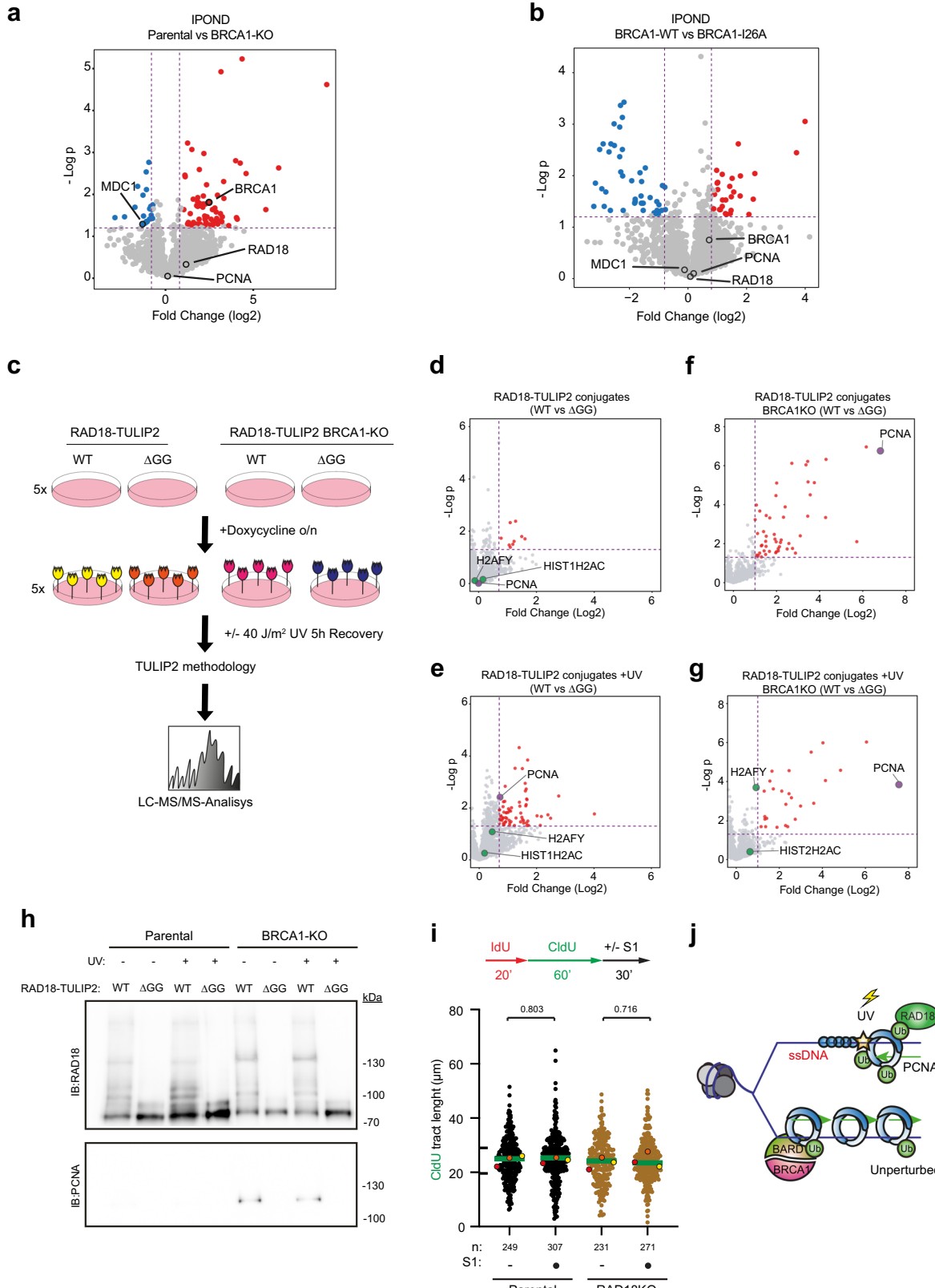

Recently, it has been proposed that the synthetic lethality between BRCA1-KO and Olaparib treatments comes from an increased replication fork speed in BRCA1 deficient cells and subsequent hyper accumulation on ssDNA gaps behind replication forks[38,39]. The accumulation of ssDNA gaps behind the fork have been also detected in PCNA-K164R mutants[36], as well as when treating BRCA1-deficient cells with mild replicative stress like HU concentrations not sufficient to produce a complete nucleotide depletion[38,42]. In this study, we show that the E3 activity BRCA1-I26A mutant show similar levels of ssDNA gaps formation as PCNA-K164R mutant and BRCA1-KO cells (Fig. 2c, d), which seems, in principle, contradictory with the E3-deficient mutant being resistant to Olaparib treatment (Fig. 1a[10,19,20]). However, Olaparib treatment does not induce PCNA ubiquitination (Supplementary Fig. 3b), thus suggesting that PARP inhibition-associated ssDNA gaps in

**Fig. 4 | RAD18 and BARD1 ubiquitinates PCNA in different pathways.** Volcano Plots depicting statistical differences in iPOND experiments comparing (**a**) Parental and BRCA1-KO cells or (**b**) BRCA1-WT or BRCA1-I26A rescued cells. *P* value corresponds to unpaired two-sided t-tests. Each dot represents a protein from Supplementary Data 4. **c** Schematic representation of RAD18-TULIP2 set up in Parental and BRCA1-KO cell lines. TULIP2 constructs were induced with 1 µg/µL doxycycline overnight. Cells were treated or not with 40 J/m² of UV light and allowed to recover for 5 h prior to cell lysis, protein purification and identification by LC-MS/MS of the RAD18-TULIP2 conjugates. RAD18-WT constructs were compared with their correspondent ΔGG constructs for conjugates identification after MS analysis. Volcano plots of RAD18-TULIP2 conjugates without UV damage (**d**) and upon UV treatment (**e**) in Parental cell lines. Volcano plots depicting RAD18-TULIP2 conjugates without UV damage (**f**) and after UV damage (**g**) in BRCA1-KO cell lines. Histones are represented in green and PCNA is shown in purple. Each dot represents a protein from Supplementary Data 5 and proteins enriched over 0.8 fold (log2) and *p* > 0.05 are displayed in red. *P* value corresponds to unpaired two-sided t-tests. **h** Immunoblotting of RAD18-TULIP2 pull down samples with and without UV damage in Parental and BRCA1-KO cells. Antibodies used are indicated. This experiment was repeated three times with similar results. **i** Formation of ssDNA gaps in unperturbed conditions in Parental, and RAD18-KO cells. Top: Scheme of the IdU/CldU pulse-labeling protocol, followed by S1 nuclease treatment. Bottom: CldU tract lengths in the indicated cell lines with and without S1 nuclease treatment. Each dot represents one fiber and the green bar represents the median. n: indicates number of fibers measured from three biological independent experiments. *p* values were calculated using two-sided Mann-Whitney tests. Colored dots indicate the median of each independent experiment. **j** Proposed model for PCNA ubiquitination in two different scenarios. During unperturbed conditions BRCA1/BARD1 ubiquitinates PCNA at low levels to promote continuous DNA synthesis. Upon replication fork barrier-inducing lesions, such as UV irradiation, RAD18 ubiquitinates PCNA at higher levels. Source data are provided as a Source Data file. **c** It has been adapted and modified from Yalçin Z, Koot D, Bezstarosti K, Salas-Lloret D, Bleijerveld OB, Boersma V, Falcone M, González-Prieto R, Altelaar M, Demmers JAA, Jacobs JJL. Ubiquitinome Profiling Reveals in Vivo UBE2D3 Targets and Implicates UBE2D3 in Protein Quality Control. Mol Cell Proteomics. 2023 Jun;22(6):100548. https://doi.org/10.1016/j.mcpro.2023.100548. which was published under CC-BY license [https://creativecommons.org/licenses/by/4.0/].

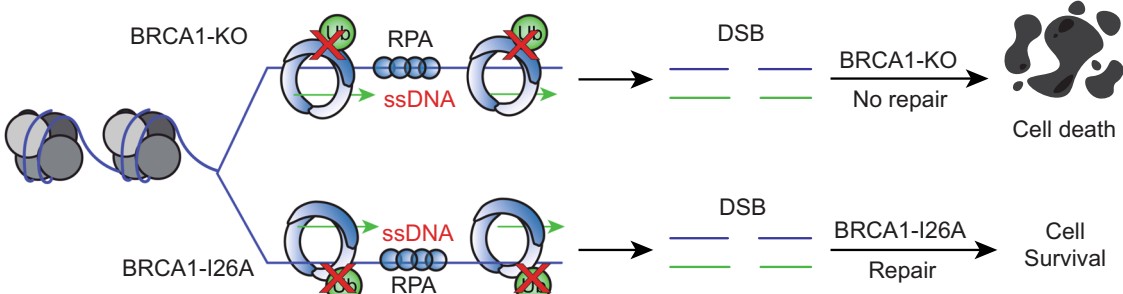

**Fig. 5 | Proposed model.** Model representing the relevance of BRCA1/BARD1 E3 activity in preventing ssDNA gap accumulation and promoting replication fork stability by PCNA ubiquitination. In the absence of BRCA1/BARD1 E3 activity, ssDNA gaps accumulate, which can eventually lead to the formation of DSBs. While in the absence of BRCA1, these DSBs would lead to cell death, in E3-deficient mutants, they can still be repaired by Homologous Recombination-mediated DSB repair.

BRCA1-deficient cells and BRCA1 E3 activity-deficiency associated gaps are formed through independent processes. We then propose a model (Fig. 5) in which upon replication stress the BRCA1/BARD1 E3 activity would promote replication fork stability and continuous, ssDNA-free, DNA synthesis. Deficiencies in the E3 activity would eventually promote the appearance of DNA DSBs. While in BRCA1-deficient cells these DSBs would lead to cell death, in the E3-deficient mutants could still be repaired through HR-mediated DSB repair. Consistently, BRCA1-KO and PCNA-K164R cells are more sensitive to HU than the BRCA1-I26A-GFP cells (Fig. 3c), as they also lack alternative resistance mechanisms such as HR-mediated DSB repair and RAD18-mediated PCNA ubiquitination pathways, respectively.

Finally, BRCA1 tumors carrying premature stop codon mutations or disruption of the RING domain are sensitive to PARP inhibitor therapies. Unfortunately, these tumors can revert by restoring the expression of a BRCA1 protein lacking the RING domain functionality but expressing the rest of the domains[11]. Alternatively, treating these patients with other chemo therapeutic agents such as HU, for which BRCA1/BARD1 E3-deficient cells are still sensitive (Fig. 3b) might open a therapeutic opportunity for cancer treatment.

## Methods
### Generation of TULIP2 and GFP-tagged constructs
BRCA1 gene was amplified and STOP codon removed from pDEST-FRT/T0-GFP-BRCA1 (Addgene #71116) by PCR using BP-tailed primers and cloned into pDNOR207 by Gateway® cloning BP reaction (Thermo Fisher Scientific). BRCA1-I26A mutation was introduced by site-directed mutagenesis. RAD18-pDNOR223 plasmid was obtained from Open Biosystems (Clone 1782) and the STOP codon was removed by site-directed mutagenesis. TULIP2 constructs were generated by Gateway® cloning LR reaction between donor plasmids containing BARD1[28], BRCA1 or RAD18 and acceptor TULIP2 plasmids[27]. For the GFP-tagged constructs, the LR reactions were performed with pLVU-GFP, gift from Lars Ittner (Addgene #24177)[59]. Primers are listed in Supplementary Table 1.

### Cell culture
293 T HEK and RPE1 cells were cultured in Dulbecco's modified Eagle's medium (DMEM) supplemented with 10 % Fetal Bovine Serum (FBS) and 100 U/mL penicillin/100 µg/mL streptomycin at 37 °C and 5% CO2 unless specifically specified. All the Parental RPE1 cells in this work were *TP53⁻/⁻*. Cells were regularly tested for mycoplasma contamination.

### TULIP2 Lentivirus production
293 T HEK cells were seeded at 30% confluency in a T175 flask containing 17 mL of DMEM + 10% FBS. After 24 h, the transfection mixture was prepared by combining lentiviral packaging plasmids 7.5 µg pMD2.G (#12259, Addgene), 11.4 µg pMDLg-RRE (#12251, Addgene), 5.4 µg pRSV-REV (#12253, Addgene) and 13.7 µg TULIP2 plasmid with 114 µL of 1 mg/mL Polyethylenimine (PEI) in 2 mL 150 mM NaCl. The mixture was vortexed and incubated 15 min at room temperature before adding to the HEK cells. Next day, culture medium was refreshed with DMEM 10% FBS. 72 h post-transfection, lentiviral suspension was harvested by filtering through a 0.45 µm syringe filter (PN4184, Pall Corporation) and kept at −20 °C for further use. Lentiviral particle concentration was determined using the HIV Type 1 p24 antigen ELISA Kit (ZeptoMetrix Corporation).

### Generation of cell lines
RPE1-hTERT *TP53⁻/⁻* and RPE1-hTERT *TP53⁻/⁻ BRCA1⁻/⁻* cells were kindly provided by Dr. Sylvie Noordermeer[60] and were seeded in 15 cm

diameter plates at 10% confluency with DMEM 10% FBS. Next day, cell culture medium was replaced with lentiviral TULIP2 constructs containing medium with 8 μg/mL polybrene. After 24 h, medium was refreshed with DMEM 10% FBS, 1% Pen/Strep. 72 h post-infection, TULIP2 positive clones were selected on puromycin.

For the BRCA1-WT-GFP and BRCA1-I26A-GFP cells, two passes after lentivirus induction, positive cells were sorted in a BD FACS Aria Illu cell sorter based on GFP intensity.

### GFP-Trap co-immunoprecipitation

GFP-trap co-immunoprecipitation were performed as in ref. 61. In brief, a confluent 15 cm dish of RPE1 $TP53^{-/-}$ $BRCA1^{-/-}$ cells rescued or not with GFP-tagged BRCA1 wild type or I26A mutant were scrapped, washed in ice-cold PBS and lysed in 1 mL of ice-cold lysis buffer (20 mM Tris pH 7.5, 150 mM NaCl, 1 mM MgCl₂, 0.5% Triton X-100, EDTA-free protease inhibitors (Roche) and 20 mM N-Ethylmaleimide (NEM)). Lysates were vortexed and incubated for 1 h at 4 °C while rotating with 500 Units of Benzonase (Millipore). Afterwards, samples were centrifuged for 1 h at 20,000 × g at 4 °C. As inputs, 20 μL of supernatants were saved per sample and the rest was incubated with 25 μL of GFP-Trap beads slurry (Chromotek) for 90 min at 4 °C while rotating. Subsequently, beads were washed three times with wash buffer (20 mM Tris pH 7.5, 150 mM NaCl, 1 mM MgCl₂, EDTA-free protease inhibitors (Roche), and 20 mM N-Ethylmaleimide (NEM)) and resuspended in LDS sample buffer 1X for immunoblotting procedures.

### Purification of TULIP2 conjugates

Following the TULIP2 methodology[27], five 15 cm diameter plates of RPE1 cells containing a TULIP2 construct, were grown up to 60% to 80% confluence. Expression of TULIP2 constructs was induced with 1 μg/mL doxycycline once 60–80% confluence was reached. For the RAD18-TULIP2 experiments, UV treatment was performed at 40 J/m² UV with 5 h recovery.

24 h after doxycycline induction, cells were washed twice with ice-cold PBS and scraped. Next, cells were spun down and collected in 5 mL ice-cold PBS, 100 μL of sample was taken as input and lysed in 200 μL SNTBS buffer (2% SDS, 1% NP-40, 50 mM TRIS pH 7.5, 150 mM NaCl). After additional centrifugation, cells were lysed in 10 mL Guanidinium buffer (6 M guanidine-HCl, 0.1 M Sodium Phosphate, 10 mM TRIS, pH 7.8) and snap frozen in liquid nitrogen. After thawing, lysates were homogenized at room temperature by sonication at 80% amplitude during 5 s using a tip sonicator (Q125 Sonicator, QSonica, Newtown, USA). Sonication was performed twice. Subsequently, protein concentration was determined by BiCinchoninic Acid (BCA) Protein Assay Reagent (Thermo Scientific). After equalization, lysates were supplemented with 5 mM β-mercaptoethanol and 50 mM Imidazole pH 8.0. 100 μL of dry nickel-nitrilotriacetic acid-agarose (Ni-NTA) beads (QIAGEN), were equilibrated with Guanidinium buffer supplemented with 5 mM β-mercaptoethanol and 50 mM Imidazole pH 8.0. Equilibrated Ni-NTA beads were added to the cell lysates and incubated overnight at 4 °C under rotation.

After lysate-beads incubation, Ni-NTA beads were transferred with Wash Buffer 1 (6 M Guanidine-HCl, 0.1 M Sodium Phosphate, 10 mM Tris, 10 mM Imidazole, 5 mM β-mercaptoethanol, 0.2 % Triton X-100, pH 7.8) to an Eppendorf LoBind tube (Eppendorf). Subsequently, beads were washed with Wash buffer 2 (8 M Urea, 0.1 M Sodium Phosphate, 10 mM Tris, 10 mM imidazole, 5 mM β-mercaptoethanol, pH 8) and transferred to a new LoBind tube with Wash buffer 3 (8 M urea, 0.1 M Sodium Phosphate, 10 mM Tris, 10 mM imidazole, 5 mM β-mercaptoethanol, pH 6.3). Ultimately, beads were washed twice with Wash buffer 4 (8 M urea, 0.1 M Sodium Phosphate, 10 mM Tris, 5 mM β-mercaptoethanol, pH 6.3). After last wash, Ni-NTA beads were resuspended in 100 μL of 7 M urea, 0.1 M NaH2PO4/Na2HPO4, 0.01 M Tris/HCl, pH 7, and 10% of the sample was taken as pull down for immunoblotting.

### Lys-C and trypsin digestion of TULIP2-purified conjugates

Ni-NTA beads were firstly digested with 500 ng recombinant Lys-C (Promega) at RT while shaking at 1400 rpm. After 5 h with Lys-C, urea buffer was diluted to <2 M by adding 50 mM ABC. A second digestion was performed o/n at 37 °C while shaking at 1400 rpm using 500 ng of sequencing-grade modified trypsin (Promega). Trypsin-digested peptides were separated from Ni-NTA beads by filtering through a 0.45 μm filter Ultrafree-MC-HV spin column (Merck-Millipore).

### iPOND

Isolation of Proteins On Nascent DNA (iPOND) technique was performed as stablished in ref. 62,with some modifications. Ten confluent 150 mm dishes of RPE1 $TP53^{-/-}$ (Parental) and RPE1 $TP53^{-/-}$ $BRCA1^{-/-}$ cells rescued or not with GFP-tagged BRCA1 wild type or I26A mutant were labeled with EdU for 15 min. Cells were fixed with 1% formaldehyde in PBS and cross-linking reaction was stopped with 1.25 M glycine for 5 min. Cells were pelleted, washed with cold PBS and incubated in permeabilization buffer (0.25% Triton X-100 in PBS) for 30 min. Cells were then washed two times with 0.5% BSA in PBS and incubated with a click reaction cocktail (1 × PBS, 10 μM biotin azide (Invitrogen), 10 mM sodium ascorbate, 2 mM CuSO₄ per 1 × 10⁸ cells) while rotating for 1–2 h. After the incubation time, cells were washed with cold 0.5% BSA and resuspended in lysis buffer (1% SDS in 50 mM Tris, pH 8.0) supplemented with aprotinin (Sigma-Aldrich) and leupeptin (Sigma-Aldrich). Lysates were sonicated (30 s constant pulse, 40 s pause at high power; total pulse time: 3 min) with a Bioruptor® Plus sonication device (Diagenode). Samples were centrifuged for 10 min at 16,000 g and the resulting lysate was diluted 1:1 (v/v) with cold PBS containing proteases inhibitors. Lysates were incubated with 100 μL of streptavidin beads (Sigma-Aldrich) slurry per 1 × 10⁸ cells for 16–20 h at 4 °C while rotating. Beads were washed with cold lysis buffer and then transferred to an Eppendorf LoBind tube (Eppendorf) with Wash buffer A (0.1% sodium deoxycholate, 1% Triton X-100, 500 mM NaCl, 1 mM EDTA, 50 mM HEPES pH 7.5). Subsequently, beads were washed with Wash buffer B (250 mM LiCl, 0.5% Triton X-100, 0.5% sodium deoxycholate, 1 mM EDTA, 10 mM TrisCl pH 8) and finally with Wash Buffer C (50 mM Tris pH 7.5, 50 mM NaCl). Samples were then prepared for mass spectrometry by washing the beads three times with 50 mM ammonium bicarbonate. Beads were digested with 250 ng of sequencing grade modified trypsin (Promega) at 37 °C while shaking at 1400 rpm overnight. Trypsin digested peptides were separated from the streptavidin beads by filtering through a 0.45 μm filter Ultrafree-MC-HV spin column (Merck-Millipore).

### Mass spectrometry sample preparation

Digested peptides were acidified by adding 2% TriFlourAcetic (TFA) acid. Subsequently, peptides were desalted and concentrated on triple-disc C18 Stage-tips as previously described[63]. Stage-tips were in-house assembled using 200 μL micro pipet tips and C18 matrix (Sigma-Aldrich). Stage-tips were activated by passing through 100 μL of methanol. Next, 100 μL of Buffer B (80% acetonitrile, 0.1% formic acid), 100 μL of Buffer A (0.1% formic acid), the acidified peptide sample, and two times 100 μL Buffer A were passed through the Stage-tip. Elution was performed twice with 25 μL of Elution buffer (32,5% acetonitrile, 0.1% formic acid solution).

Samples were vacuum dried using a SpeedVac RC10.10 (Jouan, France) and stored at −20 °C. Prior to mass spectrometry analysis, samples were reconstituted in 10 μL 0.1% formic acid and transferred to autoload vials.

### LC-MS/MS data acquisition

Mass spectrometry data was acquired either by and nanoLC Easy 1000 (Proxeon, Odense, Denmark) coupled to a Q-Exactive mass spectrometer (Thermo, Bremen, Germany) (BARD1-TULIP2 samples - Fig. 1) or Ultimate 3000 nano-gradient HPLC system (Thermo, Bremen,

Germany), coupled to an Exploris480 mass spectrometer (Thermo, Bremen, Germany) (Rest of TULIP2 samples).

For the Q-Exactive, chromatography was performed as in ref. 27 peptides were separated in an in-house packed with Reprosil-Pur C18-AQ 1.9 μm (Dr. Maisch, Ammerbuch, Germany) 20 cm analytical column in a 45 min gradient from 0% to 30% acetonitrile gradient in 0.1% Formic Acid followed of 20 min of column re-equilibration. The mass spectrometer was operated in a Data-Dependent Acquisition mode with a top-7 method and a scan range of 300–1600 *m/z*. Full-scan MS spectra were acquired at a target value of $3 \times 10^6$ and a resolution of 70,000, and the Higher-Collisional Dissociation (HCD) tandem mass spectra (MS/MS) were recorded at a target value of $1 \times 10^5$ and with a resolution of 35,000, an isolation window of 2.2 m/z, and a normalized collision energy (NCE) of 25%. The minimum AGC target was $1 \times 10^4$. The maximum MS1 and MS2 injection times were 250 and 120 ms, respectively. The precursor ion masses of scanned ions were dynamically excluded (DE) from MS/MS analysis for 20 s. Ions with charge 1, and >6, were excluded from triggering MS2 analysis.

For the Exploris480, samples were injected as in ref. 64 onto a cartridge precolumn (300 μm × 5 mm, C18 PepMap, 5 μm, 100 A) with a flow of 10 μl/min for 3 min (Thermo, Bremen, Germany) and eluted via a homemade analytical nano-HPLC column (50 cm × 75 μm; Reprosil-Pur C18-AQ 1.9 μm, 120A) (Dr. Maisch, Ammerbuch, Germany). The gradient was run from 2% to 38% solvent B (80% acetonitrile, 0.1% formic acid) in 120 min. The nano-HPLC column was drawn to a tip of ~10 μm and acted as the electrospray needle of the MS source. The temperature of the nano-HPLC column was set to 50 °C (Sonation GmbH, Biberach, Germany). The mass spectrometer was operated in data-dependent MS/MS mode for a cycle time of 3 s, with a HCD collision energy at 28 V and recording of the MS2 spectrum in the orbitrap, with a quadrupole isolation width of 1.2 Da. In the master scan (MS1) the resolution was 120,000, the scan range 350–1600, at a standard AGC target with maximum fill time of 50 ms. A lock mass correction on the background ion m/z = 445.12 was used. Precursors were dynamically excluded after $n = 1$ with an exclusion duration of 45 s, and with a precursor range of 10 ppm. Charge states 2–5 were included. For MS2 the scan range mode was set to automated, and the MS2 scan resolution was 30,000 at a normalized AGC target of 100% with a maximum fill time of 60 ms.

For the iPOND data, samples were measured using a nanoElute II LC system coupled to a timsTOF SCP mass spectrometer with an electrospray source (Bruker Daltonics). LC separations were performed on C18 HPLC column (Aurora 25 cm and 75 μm ID, IonOpticks) kept at 50 °C. Gradient elution was performed with a binary system consisting of (A) 0.1% aqueous formic acid and (B) 0.1% formic acid in CH3CN. An increasing linear gradient (v/v) was used (t (min), %B): 0, 2; 40, 17; 60, 25; 66, 37; 67, 95, followed by an equilibration step.

Mass spectrometric analysis was performed in a data independent acquisition parallel accumulation serial fragmentation (dia-PASEF) mode, with 100–1700 m/z mass range, an ion mobility range from 0.64 to 1.45 V s cm$^{-2}$, capillary voltage set to 1500 V, an accumulation and ramp time at 100 ms and the collision energy as a linear ramp from 20 eV at 1/K0 = 0.6 V s cm$^{-2}$ to 59 eV at 1/K0 = 1.6 V s cm$^{-2}$.

## Mass spectrometry data analysis

All raw data was analyzed using MaxQuant (version 1.6.7.0) as previously described[65]. Search was performed against an *in-silico* digested UniProt reference proteome for *Homo sapiens* including canonical and isoform sequences (24th January 2022). Database searches were performed according to standard settings with the following modifications: Digestion with Trypsin/P was used, allowing 4 missed cleavages. Oxidation (M), acetyl (protein N-term), phospho (S, T), and GlyGly (K) for ubiquitination sites were allowed as variable modifications with a maximum number of 3. Label-Free Quantification (LFQ)

was enabled, not allowing Fast LFQ while permitting iBAQ and matching between runs.

dia-PASEF data from the iPOND experiments was analyzed using DIA-NN (version 1.8.1)[66,67]. Using an *in-silico* predicted spectral library using UniProt reference proteome for Homo sapiens including canonical (19th January 2024), without variable modifications, enabling 1 mis cleavage with trypsin and with mass ranges of 100–1700 m/z.

Output from MaxQuant or DIA-NN data were exported and processed for statistical analysis in the Perseus computational platform version 1.6.7.0[68]. LFQ intensity values were log2 transformed and potential contaminant and proteins either identify by site or only reverse peptides were removed. Samples were grouped in experimental categories and proteins not identified in 4 out of 4 replicates (3 out of 3 for iPOND) in at least one group were removed. Missing values were imputed using normally distributed values with 0.3 width and 1.8 down shift separately for each column. After imputation, statistical analysis was performed using two-sided Student's *t* tests. Results were exported into in MS Excel 365 for a comprehensive browsing and visualization of the datasets. Volcano plots were constructed for data visualization using the VolcaNoseR web app[69] (https://huygens.science.uva.nl/VolcaNoseR2/).

## Electrophoresis and immunoblotting

Samples were separated on Novex 4–12% gradient gels (Thermo Fisher Scientific) using NuPAGE® MOPS SDS running buffer (50 mM MOPS, 50 mM Tris-base, 0.1% SDS, 1 mM EDTA pH 7.7) or in-house casted gels of different Acrylamide percentages and transferred onto Amersham Protran Premium 0.45 NC Nitrocellulose blotting membranes (GE Healthcare) using a Bolt Mini-Gel system (Thermo Fisher Scientific), which was used for both the gel electrophoresis and the protein transfer to the membrane according to vendor instructions. Membranes were stained with Ponceau-S (Sigma-Aldrich) to determine total amount of protein loaded. Next, membranes were blocked with blocking solution (8% milk, 0.1% Tween-20 in PBS) for 1 h prior to primary antibody incubation. Primary antibodies were incubated overnight and secondary antibodies for 1–2 h, except for the PCNA-Ub immunostaining, where overnight incubation with the secondary antibody was also performed. Chemiluminescence reaction was initiated with Western Bright Quantum Western blotting detection kit (Advansta-Isogen) and measured in a ChemiDocTM imaging system (BIO-RAD, Hercules, CA, USA), or an ImageQuant 800 (Cytiva, Malborough, MA, USA). Antibodies are listed in Supplementary Table 2.

## Clonogenic survival assay

RPE1 cells lines were seeded at 3000 cells/well in 6-well plates and allowed to attach overnight. TULIP2 constructs were induced with 1 μg/mL doxycycline prior treatment. Olaparib (Bio-connect) was added at different concentrations for 24 h. After treatment, medium was refreshed. Hydroxyurea (Sigma) was added at different concentrations and medium was refreshed after 16 h treatment. Subsequently, cells were allowed to grow for 10 days and fixed for 20 min in 4% paraformaldehyde (PFA) in PBS. Cells were stained with Crystal Violet 0.05% for 30 min and washed with water. Afterward, Crystal Violet was re-solubilized in methanol and O.D.595 was measured in the VICTOR X3 Multilabel Plate Reader 2030-0030 (Perkin Elmer). GraphPad Prism 10 was used for statistical analysis and the value of untreated cells was set at 100% survival.

## Immunofluorescence

RPE1 cell lines were seeded at 20% confluency on 8 mm coverslips in 12-well plates and allowed to attach overnight. 1 μg/mL doxycycline was added to required cell lines to induce TULIP2 constructs 24 h prior treatment. For RAD51 focus experiments, cells were treated with 10 Gy with 4 h recovery before fixation. For ssDNA gaps, medium was supplemented with 10 μM CldU (Sigma). After 48 h, cells were fixed with

1% PFA, 0.3% Triton X-100, 0.5% methanol after 3 h treatment. RAD51 experiments were performed using a Leica SP8 confocal microscope taking 8 frames per image. ssDNA gaps experiments were performed in a ZEISS fluorescent microscope. Fiji[70] and GraphPad Prism 10 were used for quantification and statistical analysis.

## Quantification and statistical analysis

Quantification of microscopy and immunoblotting data was performed using Fiji-ImageJ[70] and the statistical analysis was performed in GraphPad Prism 10. Statistical details of individual experiments can be found in figure legends, including the statistical test performed and definition of center and dispersion representation. For every analysis, N represents the number of values considered in the statistical analysis. For the S1-fiber analysis experiments, statistics shown in the figure have been performed according to a previous methods article by the Vindigni lab on how to perform and analyze these assays[71]. Alternatively, a statistical analysis using RM-ANOVA tests on the medians of each independent experiment is shown in (Supplementary Fig. 5).

## S1 nuclease assay

Cells were pulse-labeled with 20 μM IdU (20 min), washed twice with PBS and pulse-labeled with 200 μM CldU in the presence of 0.5 mM HU for 2 h. Cells were then washed twice with PBS and permeabilized with CSK buffer (100 mM NaCl, 10 mM MOPS pH 7, 3 mM $MgCl_2$, 300 mM Sucrose, and 0.5% Triton X-100 in water) for 8 min at RT. Permeabilized cells were treated with S1 nuclease buffer (30 mM Sodium acetate pH 4.6, 10 mM Zinc acetate, 5% glycerol, 50 mM NaCl in water) with or without 20 U/mL S1 nuclease (Invitrogen, 18001-016) for 30 min at 37 °C. Cells were then scrapped in PBS + 0.1% BSA, pelleted, and resuspended in PBS + 0.1% BSA at a final concentration of $1-2 \times 10^3$ cells/μl. 2.5 μL of cell suspension were spotted on a positively charged slide and lysed with 7.5 μL of spreading buffer (200 mM Tris-HCl pH 7.5, 50 mM EDTA, 0.5% SDS). After 8 min, slides were tilted at 45 degrees to allow the DNA to spread. Slides were then air-dried, fixed with ice-cold methanol/acetic acid (3:1) for 5 mins, air-dried, and stored at 4 °C. Slides were rehydrated with PBS, denatured with 2.5 M HCl for 1 h, washed with PBS twice, and blocked with blocking buffer (3% BSA, 0.1% Triton X-100 in PBS) for 40 min. Next, slides were incubated with primary antibody mix of mouse anti-BrdU which recognizes IdU (Becton Dickinson #347580, 1:250), and rat anti-BrdU which recognizes CldU (Abcam #6326, 1:250) diluted in blocking buffer for 2.5 h at RT in a dark humid chamber. Slides were washed three times with PBS for 5 min each and incubated with secondary antibodies anti-mouse Alexa fluor 594 and anti-rat Alexa fluor 488 (1:250, Invitrogen #A11005 and #A11006, respectively) in blocking buffer for 1 h at RT in a dark humid chamber. After washing three times with PBS and air-drying, slides were mounted with Prolong gold antifade reagent (Invitrogen, P36930) and stored at 4 °C until imaging. Images were acquired using a AF6000 Leica Fluorescence microscope equipped with a HCX PL APO 63× (NA = 1.4) oil objective. At least 200 fibers per condition were measured using the segmented line tool on ImageJ FIJI software (https://fiji.sc).

## Reporting summary

Further information on research design is available in the Nature Portfolio Reporting Summary linked to this article.

## Data availability

The mass spectrometry-based proteomics data have been deposited to the ProteomeXchange Consortium via the PRIDE[72] partner repository with the dataset identifier PXD039167 (TULIP2 data) and PXD050319 (iPOND data). Source data are provided with this paper.

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

## Acknowledgements

Authors thank Daniel Durocher and Sylvie Noordermeer for sharing the Parental and BRCA1-KO cell lines RPE1 cell lines[3,73], Anja Bielinsky for the Parental and PCNA-K164R RPE1[36] cell lines, and Peter Sterling for the Parental and RAD18-KO RPE1[55] cell lines used in this work. Research and publication of this work was funded by the EMERGIA 2020 program (EMERGIA20_00276) from the Consejería de Economía, Conocimiento, Empresas y Universidad, Junta de Andalucía, Spain to R.G.-P. Research was additionally supported by a Young Investigator Grant from the Dutch Cancer Society (KWF-KIG 11367/2017-2) and Plan Propio de Investigación VI-PP-A.Talento-IV.2 from the University of Sevilla, grants CNS2022-135216 funded by MICIU/AEI/10.13039/501100011033 and by European Union NextGenerationEU/PRTR and PID2021-122361NA-I00 by MICIU/AEI/10.13039/501100011033 and by European Union to R.G.-P. Work in the laboratory of A.C.O.V. has been supported by the European Research Council (ERC; grant 310913) and the Dutch Research Council (NWO; grant 724.016.003). N.G.-R. was supported by the EMERGIA 2021 program (EMERGIA21_00057) from the Consejería de Universidad, Investigación e Innovación, Andalusian Regional Government- Junta de Andalucía to NG-R.

## Author contributions

R.G.-P. initiated the project. D.S.-L., N.G.-R., E.S.-H., L.G.-V., C.E.-S., L.G., and R.G.-P. performed experiments in A.C.O.V., P.H., and R.G.-P. laboratories. E.S.-H., L.G.-V., and C.E.-S. contributed equally. R.G.-P., A.H.d.R., and P.A.v.V. provided mass spectrometry data acquisition support. R.G.-P., M.L.M.-M., and A.H.d.R. acquired mass spectrometry data. D.S.-L. and R.G.-P. analyzed mass spectrometry data. D.S.-L., N.G.-R., and R.G.-P. proposed and designed experiments and wrote the manuscript with input from other authors.

## Competing interests

The authors declare no competing interests.
