## [Peer Review File · Nature Communications]

BRCA1/BARD1 ubiquitinates PCNA in unperturbed conditions to promote continuous DNA synthesisREVIEWER COMMENTS

Reviewer #1 (Remarks to the Author):

The present manuscript provides evidence that BRCA1 ubiquitinates PCNA at K164 under normal conditions, and this may contribute to the suppression of single stranded DNA gap accumulation during replication. The topic is important because BRCA1 substrates have long been sought after, and ssDNA gaps have received renewed interest in recent years due to their involvement in genomic instability. The authors employ the TULIP technique, a clever approach they have previously designed, to identify PCNA as a BRCA1 substrate. The results are new and important. Next, they try to connect this ubiquitination to ssDNA gap suppression. The results here are less convincing and more work is needed to shore up this claim. Although the manuscript is very "light", with a limited number of results presented, I believe the work is important enough to deserve consideration for publication in Nature Communications.

Specific comments:

1. Using the BRCA1-TULIP fusion, the authors provide convincing data that BRCA1 ubiquitin ligase activity is required for PCNA ubiquitination under normal conditions. However, I am not convinced that the authors are able to exclude an involvement of RAD18 in this process. The experiments they provide do not directly address this issue. Instead, the authors should simply knockdown RAD18 in the BRCA1-catalytic mutant cells and perform the BRCA1-TULIP experiment. If RAD18 is not involved, its depletion should not affect the amount of PCNA pulled down under these conditions.
2. The experiments investigating DNA gaps in Figs 2C,D and Suppl Fig 2 are not showing novel results. As the authors themselves state, it was described before that both PCNA-K164R and BRCA1-KO cells accumulate gaps. The important experiment is shown in Fig3A, where the authors combine K164R mutation and BRCA1 depletion to investigate epistasis. This would indicate that indeed, BRCA1-mediated ubiquitination of PCNA suppresses DNA gap accumulation. However, this result is not convincing, in my opinion. Is this assay even able to detect a non-epistatic interaction (an increase in gap formation)? In parallel to BRCA1, the authors should also deplete BRCA2, which should presumably show an additive effect with the K164R mutation. This would be a good control to demonstrate that the assay can be used to measure epistasis. Similar epistasis experiments should be performed also under normal conditions (without HU), since the author's model is that BRCA1 ubiquitinates PCNA under normal conditions. Finally, epistasis experiments should also be performed using the CldU incorporation assay (SupplFig2) since an additive/epistatic interaction should be more easily detectable in this assay (a further increase in the signal would indicate a non-epistatic interaction).
3. I am unclear what the take-home message is for Fig 3B. It was previously shown that the K164R mutations and BRCA1-KO have HU sensitivity. There does not seem to be any novelty here. In the same vein as above, they need to perform epistasis analyses between BRCA1 and K164R.
4. The effect of BRCA1 knockout on endogenous PCNA ubiquitination in Suppl Fig 3 is not convincingly shown. The authors need to show signal quantifications with statistical analyses from multiple independent experiments.
5. The last sentence of the Abstract is awkward, in the absence of a verb.
6. The last paragraph of the Introduction section should clearly describe the results, and not simply state that "novel insights" are presented.

Reviewer #2 (Remarks to the Author):

The manuscript by Saras-Lloret et al identifies PCNA as a target for the ubiquitin ligase activity of the BRCA1-BARD1 complex. Using a direct conjugation of tagged ubiquitin to the E3 enzyme in RPE-1 cells, they show that BRCA1/BARD1 directly ubiquitinates PCNA on residue K164. They further show that this activity reduces the formation of ssDNA gaps during replication. The regulation of DNA replication and damage bypass through PCNA post-translational modifications is a long-studied subject, and the reported results can represent a significant step forward in the field. The manuscript uses clear and well-controlled experiments to identify PCNA as a BARD1 and BRCA1 ubiquitination target, but the assessment of the functional role and significance of the process is limited and not fully convincing.

Comments on the Introduction:

- The lifetime chances of cancer in BRCA1 or BARD1 mutation carriers should be expressed more precisely, not as an 'up to' value.
- It should be pointed out that the I26A ubiquitin ligase mutant differs from patient-derived RING domain mutations such as C61G, C64G in that the latter residues are indeed required for HR and for tumour suppression - and that this is likely due to the disruption of BARD1 binding.

Comments on the Results/Discussion

1. Following the finding that BRCA1 can ubiquitinate PCNA, the authors aim to (should) show that this does actually take place and that it has a physiological role. Relating to the first item, the experiment reported in Supplementary Figure 3 on endogenous PCNA ubiquitination levels is important, but it is not convincing. Why does the endogenous PCNA level vary? PCNA-Ub levels should be normalised to this. The antibodies used are not specified in the legend, nor in the methods. The PCNA-Ub signal is presumably seen with a specific antibody, but PCNA-Ub can also be visible on overdeveloped anti-PCNA blots, and these might be more reliable for quantitation.

The interpretation of the data is also problematic. The authors write that "Consistent with the olaparib and hydroxyurea survival experiments results (Figures 1A,3B), analysis by immunoblotting of PCNA levels after treatment with olaparib (Supplementary Figure 3A), showed no effect. However, treatment with hydroxyurea (Supplementary Figure 3B) caused an accumulation of PCNA ubiquitination in parental and BRCA1-WT-GFP rescued BRCA1-KO cells. This response to hydroxyurea did not occur in BRCA1-I26A-GFP rescued cells." The first sentence refers to PCNA levels, but presumably means PCNA-Ub levels. PCNA levels do indeed vary, for unknown reasons and with unknown reproducibility. PCNA-Ub levels appear much higher in the I26A overexpressing line, and decreasing with olaparib, none of this is expected. PCNA-Ub levels may also increase in response to high concentration of HU. This figure is also referred to later (line 280), apparently showing that "PCNA ubiquitination levels were lower in BRCA1 knockout cells compared to parental cells in unperturbed conditions".

Quantitation of the result of several independent repeats will help support this statement. The results should also be moved to a main figure.

This point is central to the manuscript, so it would be important to support it with a different assay or model. RAD18-dependent PCNA ubiquitination was reported in DT40 cells (PMID: 16888649). Does the elimination of BRCA1 (a mutant cell line is available) impact on PCNA ubiquitination in unperturbed cells? Alternatively, does silencing of BRCA1 reduce PCNA-Ub in human cell lines other than RPE-1?

Further to this point, the authors cite ref.51 in the discussion to say that "CRL4 works synergistically with RAD18 and CRL4 knockdowns affect the accumulation of mono-ubiquitinated PCNA in response to UV irradiation". However, both this paper from the Dutta lab and the above-mentioned one from the Sale lab clearly show a reduction of PCNA-Ub in unperturbed cells in the absence of RAD18, and in ref.51 also in the absence of CRL4. Both studies amplified the signal by knocking down/out USP1. The authors should conduct a more careful investigation of the effect of BRCA1, RAD18 and perhaps CRL4 on PCNA ubiquitination. This should include unperturbed cells, UV-treated cells and also cells exposed

to replication stress (HU) to test their earlier suggestion that RS-induced PCNA-Ub is all due to BRCA1.

2. Evidence that PCNA ubiquitination by BRCA1 has a physiological role is limited to a single epistasis experiment in Fig. 3A on gaps. All the results on Fig. 2 only show that the ubiquitin ligase function of BRCA1 has a function in replication, but does not show that this is through PCNA. Further experiments should be attempted to support this epistasis, for example performing the HU sensitivity assays of Fig. 3B on BRCA1KO-BRCA1-I126A PCNA-K164R combined mutant cells (or achieve a similar combination using siRNA).

3. PCNA can also be polyubiquitinated, which may promote fork stability or gap filling by HR. Can the authors comment on whether BRCA1 could have a role in this, e.g. by providing the first Ub, and perhaps recruiting the factors responsible for the polyubiquitination?

Minor points:

- The last sentence of the abstract is grammatically not correct.
- Intro line 60 and elsewhere: should be 'RAD51 focus formation' (foci is the plural of focus, such as 'formation of RAD51 foci')
- Fig. S2 ssDNA gap assay by immunofluorescence: representative images should also be provided
- There is no need to refer to each figure panel again in the discussion chapter.

Reviewer #3 (Remarks to the Author):

This manuscript by Salas-Lloret et al. unveils PCNA as a new substrate for the BRCA1-BARD1 dimer and proposes that this ubiquitylation is important for the maintenance of replication fork progression and to prevent the accumulation of ssDNA in unperturbed conditions. The authors provide evidence linking this activity to the DNA replication fork protection in the presence of replication stress. They argue that their data points to an ubiquitylation independent role of BRCA-BARD1 in HR in contrast to the control of DNA replication, performed through PCNA ubiquitylation. The identification of PCNA as a substrate for BRCA1-BARD1 is indeed very intriguing and constitutes a relevant finding with some potentially important implications. However, some of the results in the manuscript are not solid as they are not reproduced and/or quantified; also, some figures are contradictory. There are some very strong claims in the manuscript, such as "solving the controversy about the relevance of the ubiquitin E3 function of BRCA1-BARD1 for HR", which are not substantiated by the results. The interpretation of these results is skewed in several experiments, favoring one of the possible explanations by describing partial effects as a complete rescue of the phenotypes. In my opinion, the manuscript requires a strong revision of many experiments, including the necessary replicates that could allow a more consistent interpretation. This is especially relevant since the authors want to address a controversial point in the literature.

Here are the major points that need to be addressed by the authors:

1. The identification of PCNA as a substrate for BRCA1-BARD1 is very compelling but it should be supported by a WB analysis similar to what is shown for Rad18 in Figure 4F. In addition, an in vitro ubiquitylation assay would help understand if BRCA1-BARD1 only mono-ubiquitylates PCNA or, alternatively, chains are also formed.
2. The authors claim that BRCA1 KO cells show reduced levels of Ub-PCNA. Actually, Fig S3A does not show this, while Fig S3B does. How many times have these experiments been repeated? In order to support their claim, a quantification of several WB should be included, and the authors need to indicate how many times the experiment was repeated. This is an important point since the authors use this result to support the conclusion that BRCA1-BARD1 is the main E3 ligase for PCNA in the absence of damage.
3. Another issue arises with these experiments, since the levels of Ub-PCNA are increased by WT BRCA1 only in one experiment while they are clearly increased by the I26A mutant (Figure S3). How is

this effect explained if the mutant prevents the enzymatic activity? In general, the levels of Ub-PCNA should be shown in the same gel as the unmodified PCNA, using two different exposures. The changes in Ub-PCNA need to be quantified, normalized versus the total amount of PCNA.

4. The previous point is linked to the general lack of characterization of the different cell lines that are used. The levels and localization of BARD1 and PCNA are not shown in the absence of BRCA1 and the different rescue conditions. Similarly, we do not know what are the levels of WT BRCA1 and the I26A achieved in the rescue experiments compared to the endogenous levels in the parental cells. Anantha et al. also showed that BRCA1 I26A does not interact with BARD1 as well as the WT protein. Then, it is necessary to assess if these rescue conditions are directly comparable. How much BARD1 is in the complex with BRCA1 after the rescue with the WT and the I26A mutant? Also, are these proteins localized to replication foci as efficiently as in control cells? Cell fractionation and iPOND experiments can be carried out to assess these points. Also, the cell cycle profile in these conditions should also be analyzed to discard any possible indirect effects of BRCA1 rescue through the alterations of the cell cycle.

5. Similarly, their data suggests that RAD18 takes over in the absence of BRCA1 but there is no indication as to how this may happen. Does the absence of BRCA1 lead to changes in the levels, localization to chromatin or replisome enrichment of RAD18?

6. DNA fiber assays to measure the accumulation of ssDNA gaps have only been performed twice or once (Figure 2 and 3, respectively). These results need to be confirmed in 3 independent experiments. Statistics for these assays should be performed using the media of the n experiments performed, not counting the number of fibers as independent events. At the moment, the results are not fully reliable. In this sense Figure 2C shows the measure of CldU tracts in WT cells with two very different results.

7. There are contradictory reports as to whether the I26A mutant of BRCA1 rescues the resistance to PARP inhibitors. In addition to the work from the Morris lab, the work by Anantha et al. in 2017 (eLife) shows only a partial rescue of the sensitivity to Olaparib and a significant reduction in HR activity by the I26A mutant. Although Salas-Lloret et al favor the conclusion that the E3 activity of BRCA1 is dispensable for HR, their results are not consistent in many instances and, in others, point to similar effects as those described in Anantha (2017) or by the Morris group. While Figure 1A shows that I26A confers resistance to Olaparib, Figure S1C shows only a partial rescue of resistance by the I26A, in good agreement with the results in Anantha et al. (2017). The effect in Figure S1C is not that different from the effect of I26A in HU treatment which shows that the I26A mutant also confers partial resistance to HU (Figure 3B). Similarly, the I26A mutant partially suppresses the ssDNA gap accumulation (both in Figure S2 and Figure 2D) even if the authors claim it does not (having 3 independent experiments in the fiber assays would help get more solid results, as suggested before). These results again argue that some of the actions of BRCA1 are independent of the E3 ligase activity during fork protection.

In addition, other points should be addressed:

1. Given that BRCA1 TULIP approach is not working, I think there is too much information regarding these experiments.

2. For BARD1 TULIP experiments I would like to know what is the expression of BARD1-TULIP compared to the endogenous protein levels, and to confirm its interaction with the endogenous BRCA1. Also, the His staining for the pull-down should be shown to analyze how the purification worked.

3. Figure 2B is misleading, as it depicts ssDNA gaps as dsDNA.

4. The authors suggest that RAD18 compensates for the loss of BRCA1 and ubiquitylates PCNA. Then, the overexpression of RAD18 could be able to rescue the effect of BRCA1 KO.

RESPONSE TO REVIEWERS

We would like to thank the reviewers for their feedback which has enabled us to improve our manuscript. Changes in the manuscript are highlighted in blue. A point-by-point response to the reviewers' comments is included hereunder.

REVIEWER COMMENTS

Reviewer #1 (Remarks to the Author):

The present manuscript provides evidence that BRCA1 ubiquitinates PCNA at K164 under normal conditions, and this may contribute to the suppression of single stranded DNA gap accumulation during replication. The topic is important because BRCA1 substrates have long been sought after, and ssDNA gaps have received renewed interest in recent years due to their involvement in genomic instability. The authors employ the TULIP technique, a clever approach they have previously designed, to identify PCNA as a BRCA1 substrate. The results are new and important. Next, they try to connect this ubiquitination to ssDNA gap suppression. The results here are less convincing and more work is needed to shore up this claim. Although the manuscript is very "light", with a limited number of results presented, I believe the work is important enough to deserve consideration for publication in Nature Communications.

We appreciate the support of reviewer 1 for consideration for publication in Nature Communications, we have now performed additional work that strengthen our claims.

Specific comments:

1. Using the BRCA1-TULIP fusion, the authors provide convincing data that BRCA1 ubiquitin ligase activity is required for PCNA ubiquitination under normal conditions. However, I am not convinced that the authors are able to exclude an involvement of RAD18 in this process. The experiments they provide do not directly address this issue. Instead, the authors should simply knockdown RAD18 in the BRCA1-catalytic mutant cells and perform the BRCA1-TULIP experiment. If RAD18 is not involved, its depletion should not affect the amount of PCNA pulled down under these conditions.

We have performed the BARD1-TULIP2 in the Parental and RAD18-KO cells (see response Figure below). In this experiment, in the mass spectrometry analysis, the Parental BARD1-TULIP2 PCNA signal was significantly enriched when comparing with the BARD1-TULIP2- Δ GG, yet with a lower enrichment (0.9 log₂) than in other BARD1-TULIP2 assays previously performed in the manuscript (which are around 2 log₂). Enrichment was also lower for other targets such as Histone H2A or Macro-H2A. The absence of RAD18 caused a decrease in the PCNA signal and was even lower than the background PCNA signal in the Δ GG samples. Thus, we cannot detect BRCA1/BARD1 E3 activity with this assay.

Nevertheless, is worth noting that RAD18 is an essential gene and RAD18-KO is synthetic lethal with BRCA1-KO in a TP53-KO background (Tagliatella et al. 2021, Mol Cell). Thus, RAD18 deficiency is likely to cause a re-wiring of BRCA1 activity, not necessarily involving the E3 activity, that may cause the decrease in the PCNA signal in MS analysis of TULIP2 experiments. Taking into account that the BARD1-TULIP2 expression levels are negligible compared to endogenous, sequestering the BARD1-TULIP2 constructs in other processes not necessarily involving the E3 activity may reduce the PCNA signal enough to not being significantly enriched compared with the background binding of PCNA to the beads.

There is another example in this manuscript where the signal was not distinguishable from the background binding, namely, the RAD18-TULIP2 activity towards PCNA in unperturbed conditions in Parental cells (Figure 4B). However, it is unlikely that RAD18 has null activity in unperturbed conditions towards PCNA, thus we overexposed after overnight incubation with secondary antibody the samples in Figure 4F. Here we could detect RAD18-TULIP2 signal towards PCNA in unperturbed conditions although we were not able in the MS analysis. Unfortunately, BARD1-TULIP2-PCNA conjugates do not transfer as well in immunoblotting as RAD18-TULIP2-PCNA conjugates (see page 9 of this rebuttal).

(A-B) Volcano plots depicting statistical differences between BARD1-TULIP2 conjugates and BARD1-TULIP2-ΔGG control samples in Parental (A) and RAD18-KO cells. (C) Samples from Figure 4F in the manuscript where the anti PCNA blot is overexposed.

Nevertheless, we have now investigated the formation of ssDNA gaps in unperturbed conditions in RAD18-KO cells (new Figure 4H). While the absence of BRCA1 ubiquitin E3 catalytic activity causes the formation of ssDNA gaps similarly to PCNA-K164R mutant in unperturbed conditions, the absence of RAD18 has no effect in ssDNA gap formation. We believe that this new data is sufficient evidence that the replication-associated ubiquitination of PCNA in unperturbed conditions is dependent on BRCA1 E3 activity and not on RAD18.

2. The experiments investigating DNA gaps in Figs 2C,D and Suppl Fig 2 are not showing novel results. As the authors themselves state, it was described before that both PCNA-K164R and BRCA1-KO cells accumulate gaps. The important experiment is shown in Fig3A, where the authors combine K164R mutation and BRCA1 depletion to investigate epistasis. This would indicate that indeed, BRCA1-mediated ubiquitination of PCNA suppresses DNA gap accumulation. However, this result is not convincing, in my opinion. Is this assay even able to detect a non-epistatic interaction (an increase in gap formation)? In parallel to BRCA1, the authors should also deplete BRCA2, which should presumably show an additive effect with the K164R mutation. This would be a good control to demonstrate that the assay can be used to measure epistasis. Similar epistasis experiments should be performed also under normal conditions (without HU), since the author's model is that BRCA1 ubiquitinates PCNA under normal conditions. Finally, epistasis experiments should also be performed using the CldU incorporation assay (SupplFig2) since an additive/epistatic interaction should be more easily detectable in this assay (a further increase in the signal would indicate a non-epistatic interaction).

We agree with this reviewer that the results showing the accumulation of ssDNA in BRCA1-KO, and PCNA K164R cells are not novel. However, these mutants were included as positive controls for the BRCA1 Wild type- and I26A mutants-rescued cells.

It is the effect of the absence of BRCA1/BARD1 E3 activity on ssDNA gap formation what we consider novel results.

Regarding the epistasis, we have now performed these experiments in unperturbed conditions in triplicate (new Figure 3A). The assay is indeed able to detect a non-epistatic interaction as previously shown by inhibiting PolQ in BRCA1-KO cells (Schrempf et al. 2022 Cell Reports). As we presumed that BRCA1-KO would be also epistatic over BRCA2-KO, we performed the ssDNA fiber assay in Parental or PCNA-K164R cells after inhibition of PolQ with the ART-558 inhibitor (New figure 3B). Here, we can observe an additional shortening of DNA tracts, indicating increased formation of gaps, in PCNA-K164R cells upon inhibition of PolQ, thus detecting a non-epistatic interaction.

As the S1 nuclease fiber assay is the standard in the field for detecting replication-associated ssDNA gaps, we respectfully disagree with this reviewer about repeating these experiments with the CldU incorporation assay. The CldU incorporation assay is unable of distinguishing replication-associated ssDNA gaps from other sources of ssDNA such as DNA double strand break resection, degradation of stalled replication forks or Global Genome-Nucleotide Excision Repair (GG-NER), and, for this reason, these results were included as supplemental. We have mentioned this caveat of the CldU foci assay in the manuscript text:

“... However, this assay cannot distinguish replication induced ssDNA gaps from ssDNA originating from other sources, such as DSB resection or other DNA repair intermediates....”

3. I am unclear what the take-home message is for Fig 3B. It was previously shown that the K164R mutations and BRCA1-KO have HU sensitivity. There does not seem to be any novelty here. In the same vein as above, they need to perform epistasis analyses between BRCA1 and K164R.

We would like to clarify that the K164R and BRCA1-KO mutants were included as positive controls. What we wanted to investigate here is the sensitivity of the BRCA1-I26A mutant. To clarify this point we have introduced the experiment with the following sentences:

“As lack of BRCA1 E3 activity causes the accumulation of replication associated ssDNA gaps, BRCA1-I26A mutant should be sensitive to replicative stress. Thus, we decided to test the sensitivity...”

4. The effect of BRCA1 knockout on endogenous PCNA ubiquitination in Suppl Fig 3 is not convincingly shown. The authors need to show signal quantifications with statistical analyses from multiple independent experiments.

We have now included an alternative blot image from another experimental repeat plus western blot quantifications from 4 independent experiments in these figures (new supplemental figures 3B-C) normalized to untreated conditions to measure the response to Olaparib and Hydroxyurea. Additionally, we have included another quantification of the basal PCNA Ubiquitination levels in unperturbed conditions of the same experiments (Supplementary Figures 3D-E).

5. The last sentence of the Abstract is awkward, in the absence of a verb.

The last sentence has been re-phrased to:

“These results address the controversy about the function of BRCA1/BARD1 E3 activity in Homologous Recombination.”

6. The last paragraph of the Introduction section should clearly describe the results, and not simply state that “novel insights” are presented.

We appreciate this suggestion, the last sentence of the introduction now states:
“Combined, our data showed that BRCA1/BARD1 promotes PCNA ubiquitination in unperturbed conditions to facilitate continuous DNA synthesis and enabled us to solve the controversy in the field about the relevance of the ubiquitin E3 function of BRCA1/BARD1 for homologous recombination.”

Reviewer #2 (Remarks to the Author):

The manuscript by Saras-Lloret et al identifies PCNA as a target for the ubiquitin ligase activity of the BRCA1-BARD1 complex. Using a direct conjugation of tagged ubiquitin to the E3 enzyme in RPE-1 cells, they show that BRCA1/BARD1 directly ubiquitinates PCNA on residue K164. They further show that this activity reduces the formation of ssDNA gaps during replication. The regulation of DNA replication and damage bypass through PCNA post-translational modifications is a long-studied subject, and the reported results can represent a significant step forward in the field. The manuscript uses clear and well-controlled experiments to identify PCNA as a BARD1 and BRCA1 ubiquitination target, but the assessment of the functional role and significance of the process is limited and not fully convincing.

We thank this reviewer for his/her comments and provide new data that address the functional role of PCNA ubiquitination during DNA replication by BRCA1/BARD1 in a more convincing manner. We have now repeated all the fiber assays in unchallenged conditions in triplicate, we have shown that the S1-fiber epistasis assay has window to detect non-epistatic interactions and we have shown that the absence of RAD18 does not cause ssDNA gap accumulation.

Comments on the Introduction:

- The lifetime chances of cancer in BRCA1 or BARD1 mutation carriers should be expressed more precisely, not as an ‘up to’ value.

The exact value of lifetime probability differs depending on the study and the cohort. For this reason, we prefer to opt for an “up to” value in the introduction.

- It should be pointed out that the I26A ubiquitin ligase mutant differs from patient-derived RING domain mutations such as C61G, C64G in that the latter residues are indeed required for HR and for tumour suppression - and that this is likely due to the disruption of BARD1 binding.

We have included this information in the introduction:

“...as patient-derived BRCA1 RING domain mutations such as C61G or C64G which are required for tumor suppression and HR are likely due to the disruption of the interaction of BRCA1 with BARD1¹¹. Nevertheless...”

Comments on the Results/Discussion

1. Following the finding that BRCA1 can ubiquitinate PCNA, the authors aim to (should) show that this does actually take place and that it has a physiological role. Relating to the first item, the experiment reported in Supplementary Figure 3 on endogenous PCNA ubiquitination levels is important, but it is not convincing. Why does the endogenous PCNA level vary? PCNA-Ub levels should be normalised to this. The antibodies used are not specified in the legend, nor in the methods. The PCNA-Ub signal is presumably seen with a specific antibody, but PCNA-Ub can also be visible on overdeveloped anti-PCNA blots, and these might be more reliable for quantitation.

We have now included quantifications of 4 independent experiments were ubiquitinated PCNA are normalized to total PCNA levels (Supplementary Figures 3B-C). The variation on total PCNA levels were in part due to uneven loading. Note that we have included an alternative repeat where lanes are better equalized. The antibody used is specified in supplementary table S2, and, as pointed out by this reviewer, PCNA-Ub can be seen by over developed anti-PCNA blots, which is indeed the case of our figures.

However, in order to detect by immunoblotting the endogenous ubiquitinated PCNA in unperturbed conditions we need to cut the membrane over the 30 kDa size marker to avoid unmodified PCNA to sequester the antibody and incubate it independently overnight with the primary antibody and then again overnight with the secondary antibody. Then the blots need to be overexposed.

The interpretation of the data is also problematic. The authors write that “Consistent with the olaparib and hydroxyurea survival experiments results (Figures 1A,3B), analysis by immunoblotting of PCNA levels after treatment with olaparib (Supplementary Figure 3A), showed no effect. However, treatment with hydroxyurea (Supplementary Figure 3B) caused an accumulation of PCNA ubiquitination in parental and BRCA1-WT-GFP rescued BRCA1-KO cells. This response to hydroxyurea did not occur in BRCA1-I26A-GFP rescued cells.” The first sentence refers to PCNA levels, but presumably means PCNA-Ub levels.

Indeed we meant PCNA Ubiquitination levels, we have corrected this in the manuscript.

PCNA levels do indeed vary, for unknown reasons and with unknown reproducibility. PCNA-Ub levels appear much higher in the I26A overexpressing line, and decreasing with olaparib, none of this is expected. PCNA-Ub levels may also increase in response to high concentration of HU.

This figure is also referred to later (line 280), apparently showing that “PCNA ubiquitination levels were lower in BRCA1 knockout cells compared to parental cells in unperturbed conditions”. Quantitation of the result of several independent repeats will help support this statement. The results should also be moved to a main figure.

As previously mentioned for reviewer 1, quantifications of 4 independent experiments have been included (Supplementary Figures 3D-E). We still think that this figure fits better as supplemental, but will move it to a main figure on editorial consideration.

This point is central to the manuscript, so it would be important to support it with a different assay or model. RAD18-dependent PCNA ubiquitination was reported in DT40 cells (PMID: 16888649). Does the elimination of BRCA1 (a mutant cell line is available) impact on PCNA ubiquitination in unperturbed cells? Alternatively, does silencing of BRCA1 reduce PCNA-Ub in human cell lines other than RPE-1?

As above mentioned, and after quantification, on average, the elimination of BRCA1 causes a decrease of basal PCNA ubiquitination levels between 55% (Supplemental figure 3D) and 75% (Supplemental figure 3E). The silencing of BRCA1 has been seen to reduce the levels of PCNA-Ub also in MCF7, U2OS (human) and MEF (mouse) cells (Tian et al. 2013 PNAS).

Further to this point, the authors cite ref.51 in the discussion to say that “CRL4 works synergistically with RAD18 and CRL4 knockdowns affect the accumulation of mono-ubiquitinated PCNA in response to UV irradiation”. However, both this paper from the Dutta lab and the above-mentioned one from the Sale lab clearly show a reduction of PCNA-Ub in unperturbed cells in the absence of RAD18, and in ref.51 also in the absence of CRL4. Both studies amplified the signal by knocking down/out USP1. The authors should conduct a more careful investigation of the effect of BRCA1, RAD18 and perhaps CRL4 on PCNA ubiquitination. This should include unperturbed cells, UV-treated cells and also cells exposed to replication stress (HU) to test their earlier suggestion that RS-induced PCNA-Ub is all due to BRCA1.

We have already shown that the response to UV-light in terms of PCNA ubiquitination does not depend on BRCA1 but on RAD18 (Supplementary Figure 4A-B). In our hands, RAD18-KO also produces a big reduction (around 75% reduction) in ubiquitinated PCNA levels in unperturbed conditions and these levels still increase in response to mild hydroxyurea treatment (which is the relevant in this article). Two repeats are here included.

Noteworthy, both BRCA1 and RAD18 are essential genes and KOs are only viable after additional knockout of TP53. Therefore, the effect of the knockouts can be indirect effects. The advantage of our TULIP2 approach is that enables us to monitor the ubiquitination activity of the E3s in a direct manner in a scenario where all the other E3s are present. TULIP2 constructs are expressed below endogenous levels which reduces the probability of over-expression artefacts. Importantly, although both RAD18 and BRCA1 knockout reduces PCNA ubiquitination levels, only BRCA1 E3 activity deficiency and not RAD18 deficiency recapitulates the PCNA K164R mutant phenotype regarding replication-associated ssDNA gap formation.

2. Evidence that PCNA ubiquitination by BRCA1 has a physiological role is limited to a single epistasis experiment in Fig. 3A on gaps. All the results on Fig. 2 only show that the ubiquitin ligase function of BRCA1 has a function in replication, but does not show that this is through PCNA. Further experiments should be attempted to support this epistasis, for example performing the HU sensitivity assays of Fig. 3B on BRCA1KO-BRCA1-I126A PCNA-K164R combined mutant cells (or achieve a similar combination using siRNA).

As suggested, further experiments have been performed to support the epistasis. Namely, the epistasis experiments have been performed in unperturbed conditions (new Figure 3A). Additionally, we have shown that inhibition of PolQ produces

additional gap formation, which indicates that non-epistatic interactions can be detected by this assay.

Regarding the HU sensitivity assay, we introduced BRCA1-WT-GFP and BRCA1-I26A-GFP constructs resistant to an shRNA against BRCA1 in the PCNA-K164R cells and then treated the cells with a Control or BRCA1-targetting shRNA. As expected (Thakar et al. 2020, knockdown of BRCA1 in PCNA-K164R cells caused an additional sensitivity to HU. However, when comparing the rescued cells, these were both more sensitive than the PCNA-K164R cells and no additional effect of the BRCA1 knockdown was observed (see below data from 3 independent experiments). Thus, this experiment was not conclusive.

3. PCNA can also be polyubiquitinated, which may promote fork stability or gap filling by HR. Can the authors comment on whether BRCA1 could have a role in this, e.g. by providing the first Ub, and perhaps recruiting the factors responsible for the polyubiquitination?

In this regard, we can only speculate. Our experimental evidence only shows an effect on PCNA monoubiquitination. We have also analyzed by mass spectrometry the colPs shown in new Supplementary Figure 1B and observe a decrease in the interaction with the ubiquitin E2 UBE2N in the I26A mutant. UBE2N is also the E2 for HLTF and SHPRH, E3 enzymes which can promote PCNA polyubiquitination that results in fork

reversal to promote fork stability. Thus, indeed, it is a possibility that BRCA1 could perform the first ubiquitination and then extended by HLTF/SHPRH if fork reversal is required. This is an interesting hypothesis which would fall out of the scope of the manuscript, but worth testing in a follow-up project.

Minor points:

- The last sentence of the abstract is grammatically not correct.

We have corrected this sentence.

- Intro line 60 and elsewhere: should be 'RAD51 focus formation' (foci is the plural of focus, such as 'formation of RAD51 foci')

We are aware of the focus-foci singular-plural correspondence. After IR, normally multiple RAD51 foci are formed, and a single RAD51 focus is considered a negative result. For this reason we mention RAD51 foci, because they are multiple.

- Fig. S2 ssDNA gap assay by immunofluorescence: representative images should also be provided

Representative images have been included.

- There is no need to refer to each figure panel again in the discussion chapter.

We think that for clarity, there is no harm in referring again. We prefer to keep the referring but would remove it on editorial consideration.

Reviewer #3 (Remarks to the Author):

This manuscript by Salas-Lloret et al. unveils PCNA as a new substrate for the BRCA1-BARD1 dimer and proposes that this ubiquitylation is important for the maintenance of replication fork progression and to prevent the accumulation of ssDNA in unperturbed conditions. The authors provide evidence linking this activity to the DNA replication fork protection in the presence of replication stress. They argue that their data points to an ubiquitylation independent role of BRCA-BARD1 in HR in contrast to the control of DNA replication, performed through PCNA ubiquitylation. The identification of PCNA as a substrate for BRCA1-BARD1 is indeed very intriguing and constitutes a relevant finding with some potentially important implications. However, some of the results in the manuscript are not solid as they are not reproduced and/or quantified; also, some figures are contradictory. There are some very strong claims in the manuscript, such as "solving the controversy about the relevance of the ubiquitin E3 function of BRCA1-BARD1 for HR", which are not substantiated by the results. The interpretation of these results is skewed in several experiments, favoring one of the possible explanations by describing partial effects as a complete rescue of the phenotypes. In my opinion, the manuscript requires a strong revision of many experiments, including the necessary replicates that could allow a more consistent interpretation. This is especially relevant since the authors want to address a controversial point in the literature.

We thank this reviewer for recognizing the relevance of our findings in the field. We are confident that the newly included data after revision make our results more solid and their interpretation more convincing.

Here are the major points that need to be addressed by the authors:

1. The identification of PCNA as a substrate for BRCA1-BARD1 is very compelling but it should be supported by a WB analysis similar to what is shown for Rad18 in Figure 4F. In addition, an *in vitro* ubiquitylation assay would help understand if BRCA1-BARD1 only mono-ubiquitylates PCNA or, alternatively, chains are also formed.

Regarding the WB similar to what is shown for RAD18, note that the conjugation of the TULIP construct to the substrate to PCNA forms a large multi-branched polypeptide. This branching hinders gel migration and transfer in Western Blot experiments, which in the case of RAD18-TULIP2-PCNA branched polypeptide is compensated by a log₂ fold increase enrichment of 8-10 times in a BRCA-KO background. However, this enrichment is much lower for BARD1-TULIP2-PCNA conjugate and, additionally, BARD1 is a bigger protein than RAD18, which is also not beneficial for WB transfer. Thus, we have not been able to visualize it by WB analysis in a similar way that in the case of RAD18 in the BRCA1-KO cells.

Nevertheless, by scaling up the experiment and overexposing the blots, BARD1-TULIP2-PCNA ubiquitination can be detected (See included image).

We respectfully disagree with the necessity of performing *in vitro* ubiquitination assays would be indicative of what happens *in vivo*. As several ubiquitin E3s ubiquitinate PCNA in alternative scenarios, a full replication fork with all its components would have to be reconstituted *in vitro*, also bearing specific DNA structures yet to be described, and probably specific post-translational modifications, thus performing this assay properly would be incredibly challenging and out of the scope of this manuscript.

Besides, the ubiquitin chain linkage, length and formation efficiency depend highly in the ubiquitin E2 employed in the assay.

2. The authors claim that BRCA1 KO cells show reduced levels of Ub-PCNA. Actually, Fig S3A does not show this, while Fig S3B does. How many times have these

experiments been repeated? In order to support their claim, a quantification of several WB should be included, and the authors need to indicate how many times the experiment was repeated. This is an important point since the authors use this result to support the conclusion that BRCA1-BARD1 is the main E3 ligase for PCNA in the absence of damage.

As above mentioned for both reviewers 1 and 2, we have included now an alternative blot for S3A (now S3B) and quantifications for 4 independent experiments both for S3A and S3B (Now S3B-E)

3. Another issue arises with these experiments, since the levels of Ub-PCNA are increased by WT BRCA1 only in one experiment while they are clearly increased by the I26A mutant (Figure S3). How is this effect explained if the mutant prevents the enzymatic activity? In general, the levels of Ub-PCNA should be shown in the same gel as the unmodified PCNA, using two different exposures. The changes in Ub-PCNA need to be quantified, normalized versus the total amount of PCNA.

As explained in the response to reviewer 1. Indeed, the levels of Ub-PCNA are from the same gel using the same antibody. In our case, to be able to detect endogenous PCNA ubiquitination in unperturbed conditions, we need to cut the membrane after gel transfer over the 30kDa size mark and incubate independently overnight with secondary antibody to avoid the unmodified PCNA signal to mask the modified PCNA signal.

After quantification, we can observe that, on average, basal PCNA ubiquitination levels in the BRCA1 WT-rescued cells are higher than in the I26A mutant. Both for the WT- and I26A-rescued cells, the basal levels are higher than in the BRCA1-KO cells. This can be explained because the absence BRCA1 can lead to replication fork collapse which can reflect in reduced levels of PCNA-Ub.

4. The previous point is linked to the general lack of characterization of the different cell lines that are used. The levels and localization of BARD1 and PCNA are not shown in the absence of BRCA1 and the different rescue conditions. Similarly, we do not know what are the levels of WT BRCA1 and the I26A achieved in the rescue experiments compared to the endogenous levels in the parental cells. Anantha et al. also showed that BRCA1 I26A does not interact with BARD1 as well as the WT protein. Then, it is necessary to assess if these rescue conditions are directly comparable. How much BARD1 is in the complex with BRCA1 after the rescue with the WT and the I26A mutant?

We have included new Supplementary Figures 1A-B. Here we show that the WT and I26A BRCA1-GFP construct are expressed at similar levels to the parental cells and that the interaction with BARD1 is not affected by the I26A mutation. Moreover, BARD1 levels are not affected by the absence of BRCA1 in our RPE1 cells (Figure 1E).

Also, are these proteins localized to replication foci as efficiently as in control cells? Cell fractionation and iPOND experiments can be carried out to assess these points. Also, the cell cycle profile in these conditions should also be analyzed to discard any possible indirect effects of BRCA1 rescue through the alterations of the cell cycle.

We have performed several iPOND experiments trying to measure recruitment of BRCA1/BARD1 to replication forks by immunoblotting. However, BRCA1/BARD1 are not detectable by iPOND- Western Blotting (see communication by David Cortez in Xu

et al. eLife. 2017; 6: e30523. "According to him (David Cortez), iPOND-immuno-blot is limited to detect small proteins that have very good antibodies available, such as RPA, Histone H3 and PCNA. For large proteins, the antigen epitopes may be damaged by formaldehyde crosslinking, so that they will be difficult to detect by immunoblotting. He said "I don't think western blots after iPOND for BRCA1, 53BP1, SLX4, and Mus81 will be useful or practical. Doubtful that anyone could do them successfully."

We have also checked other iPOND-MS screens in unperturbed conditions from the Fernandez-Capetillo laboratory (Lopez-Contreras et al. 2013, Cell Reports) of David Cortez laboratory (Dungrawala et al. 2015 Molecular Cell), and BRCA1/BARD1 were neither detected in unperturbed conditions. Nevertheless, We moved to an iPOND- LC-MS/MS approach using a DIA on a very sensitive timsTOF-SCP mass spectrometer. Here, we could detect, as expected, BRCA1 enriched at replication forks in the Parental cells Vs the BRCA1-KO cells, but no differences at RAD18 levels at the fork below Figure panel A). In the case of BRCA1-WT and BRCA1-I26A, neither the presence of RAD18 nor BRCA1 was affected by BRCA1 E3 activity at replication forks Below Figure panel B).

Statistical analysis of identified proteins in IPOND experiments comparing (A) Parental vs BRCA1-KO samples and (B) BRCA1-WT-rescued vs BRCA1-I26A-rescued BRCA1-KO cells. Analysis corresponds to 3 independent biological replicates.

We think that these IPOND experiments do not add to the message of the manuscript. Thus, we have opted for including in them in the rebuttal and not in the article, but we would add them as supplementary on request.

Regarding possible indirect effects of BRCA1 rescue on the cell cycle. The S1-fiber assays only measure gaps in advancing replication forks, thus is not affected by differences in cell cycle distribution.

5. Similarly, their data suggests that RAD18 takes over in the absence of BRCA1 but there is no indication as to how this may happen. Does the absence of BRCA1 lead to changes in the levels, localization to chromatin or replisome enrichment of RAD18?

We agree with this reviewer that understanding how this happens mechanistically is very intriguing and worth studying in a follow-up project. However, the purpose of this experiment was to serve as a control to exclude the possibility that the decrease in PCNA ubiquitination and subsequent accumulation of ssDNA gaps was not due to a reduced ubiquitination activity of RAD18 towards PCNA.

The data showing that the lack BRCA1 E3 activity but not the lack of RAD18 phenocopies PCNA-K164R mutant regarding ssDNA gap formation and increased RAD18-TULIP2 activity towards PCNA in the absence of BRCA1 clearly exclude this possibility.

6. DNA fiber assays to measure the accumulation of ssDNA gaps have only been performed twice or once (Figure 2 and 3, respectively). These results need to be confirmed in 3 independent experiments. Statistics for these assays should be performed using the media of the n experiments performed, not counting the number of fibers as independent events. At the moment, the results are not fully reliable. In this sense Figure 2C shows the measure of CldU tracts in WT cells with two very different results.

We have now included data from 3 independent biological repeats in the main figures to strengthen our analysis. However, it is important to note that in the published literature, only two independent repeats of DNA fiber assays are frequently shown (see for instance Cong et al., 2021, Nayak et al., 2020, Belan et al., 2022). Regarding the statistics in DNA fiber assays, the use of the Mann-Whitney test is the norm. In this sense, we refer to a method paper from the Vindigni lab (Quinet et al., 2017, Methods in Enzymology), where they stated the following:

“Graphical representations with column bars corresponding to averages are not typically used because they do not adequately address the heterogeneity of the fibers. Data are frequently plotted as scatter dot plots, or in box and whiskers plots. Statistical significance between two groups is normally assessed using the Mann–Whitney U test (unpaired and nonparametric).”

Therefore, we have performed accordingly. Nevertheless, if we repeat the analysis as this reviewer suggests, similar results are achieved, although due to the reduction in the N, p-values are, as expected, less significant.

We agree that the median of the CldU tract length slightly differ in the parental cell lines between Figure 2C and 2D, but it is important to mention that these two parental RPE1 cell lines come from different laboratories.

Nevertheless, we consistently observed the same trend in each individual experiment, in terms of sensitivity to S1 nuclease digestion. Furthermore, each graph only represent data obtained within the same experiment, using cell lines derived from the same parental. Hence, we strongly believe that the results shown in these assays are fully reliable.

7. There are contradictory reports as to whether the I26A mutant of BRCA1 rescues the resistance to PARP inhibitors. In addition to the work from the Morris lab, the work by Anantha et al. in 2017 (eLife) shows only a partial rescue of the sensitivity to Olaparib and a significant reduction in HR activity by the I26A mutant. Although Salas-Lloret et al favor the conclusion that the E3 activity of BRCA1 is dispensable for HR, their results are not consistent in many instances and, in others, point to similar effects as those described in Anantha (2017) or by the Morris group. While Figure 1A shows that I26A confers resistance to Olaparib, Figure S1C shows only a partial rescue of resistance by the I26A, in good agreement with the results in Anantha et al. (2017).

The effect in Figure S1C is not that different from the effect of I26A in HU treatment which shows that the I26A mutant also confers partial resistance to HU (Figure 3B). Similarly, the I26A mutant partially suppresses the ssDNA gap accumulation (both

in Figure S2 and Figure 2D) even if the authors claim it does not (having 3 independent experiments in the fiber assays would help get more solid results, as suggested before). These results again argue that some of the actions of BRCA1 are independent of the E3 ligase activity during fork protection.

Note that, while the expression levels of the BRCA1-GFP constructs are equivalent (Supplementary Figure 1A), in the case of the levels of the BRCA1-TULIP2 constructs, these are lower in the case of the I26A mutant, which explain the apparent contradiction of full rescue in Figure 1 and almost-full rescue in Supplementary Figure 1. Nevertheless, all the functional assays have been performed with the BRCA1-GFP rescued cells, which fully rescues the resistance to Olaparib.

In the new Figure 2C, where the assay has been performed in unperturbed conditions in triplicate, the I26A mutant phenocopies the ssDNA gap formation present in the BRCA1-KO mutant. Apparent partial rescue of the I26A mutant in mild hydroxyurea conditions (now Supplementary Figure 2B) can be explained because BRCA1-KO cells have DNA replication less restrained (higher fork speed) in response to HU in comparison to the I26A mutant cells.

We have included the three independent repeats requested in the unperturbed conditions experiments.

In addition, other points should be addressed:

1. Given that BRCA1 TULIP approach is not working, I think there is too much information regarding these experiments.

We think that these experiments should be included, otherwise, they would be obvious experiments to request.

2. For BARD1 TULIP experiments I would like to know what is the expression of BARD1-TULIP compared to the endogenous protein levels, and to confirm its interaction with the endogenous BRCA1. Also, the His staining for the pull-down should be shown to analyze how the purification worked.

We do not have a quantification, but, as shown in Figure 1E input samples, the expression levels of the BARD1-TULIP2 construct are negligible respect to the endogenous BARD1 levels. These are very efficiently enriched if we compare the input with the pulldown samples. We did not perform WB using an anti-HIS antibody as we had a more-specific BARD1 antibody available. These samples were completely digested with LysC/Trypsin and analyzed by MS, thus, it is not possible to run them again to perform the anti-HIS antibody western blot in the same samples.

3. Figure 2B is misleading, as it depicts ssDNA gaps as dsDNA.

We thank this reviewer to pointing this out and we modified the figure accordingly. We think the new version is not misleading.

4. The authors suggest that RAD18 compensates for the loss of BRCA1 and ubiquitylates PCNA. Then, the overexpression of RAD18 could be able to rescue the effect of BRCA1 KO.

In the absence of BRCA1, RAD18 increases its activity towards PCNA in a log2 fold of 8, yet it is not able to inhibit the ssDNA formation. Besides, BRCA1 has many other

functions regardless the ubiquitin E3 activity. Hence, we consider unlikely that the overexpression of RAD18 is able to rescue the effect of BRCA1-KO.

REVIEWER COMMENTS

Reviewer #1 (Remarks to the Author):

The revised manuscript addressed my comments in a satisfactory manner. In particular, the authors investigated the formation of ssDNA gaps under unperturbed conditions in RAD18-knockout cells, performed epistasis experiments, and clarified the ubiquitination of endogenous PCNA. In addition, they clarified a number of issues I had raised. Overall, in my opinion, the manuscript is suitable for publication in Nature Communications.

Reviewer #2 (Remarks to the Author):

The authors have addressed most of the referees' comments and made significant improvements to the manuscript. My general opinion is that these improvements do not quite go far enough, and some of the new data and arguments, even if contradictory, should be included in the manuscript.

Specific points:

1. I welcome the addition to the introduction, but suggest the following modification:

“...as THE EFFECTS OF patient-derived BRCA1 RING domain mutations such as C61G or C64G which are required for tumor suppression and HR are likely due to the disruption of the interaction of BRCA1 with BARD1”

2. Regarding the new quantified data in figure S3, it is surprising that PCNA ubiquitylation is apparently induced by HU in BRCA1 mutant cells, as much or even more than in the rescued ones. (Fig. S3C), and more than in those expressing the I26A mutant. The remark that “yet these levels were lower than in the other three cell lines” (which is not supported by statistics, and should reference Fig. S3E) provides little explanation for this result that does not fit with the model. It is also a problem that no statistics are provided in any part of Fig. S3. This set of experiments should be performed in a sufficient number of variations (e.g. including other drugs) to provide a convincing dataset for main figures with significant differences showing that PCNA ubiquitylation in some specific settings is dependent on BRCA1 ubiquitin ligase activity - otherwise it must be made clear that this has not been proven.

3. The authors provide experimental data in the rebuttal showing a 75% reduction of PCNA-Ub in unperturbed cells upon knockout of RAD18. I presume these are RPE-1 p53^{-/-} cells, but these data do not seem to be in the manuscript. At the same time, they could not find RAD18-TULIP-PCNA conjugates in unperturbed cells, only upon UV treatment. This is a clear discrepancy, which could be scientifically interesting. The data provided in the revision should be included in the paper and discussed.

In the revision the authors argue that the KO of p53 could interfere with the results, but I do not really see the argument. Aren't all tested cell lines RPE-1 p53^{-/-}? The 'parental' cell line should be precisely specified in the methods and in the figure legends.

4. The new epistasis experiment in Fig. 3A is useful, as is the inhibition of pol theta in Fig. 3B. However, I must echo reviewer 3 in drawing attention to the bad practice that fibres from the same experiment are treated as individual data points for the purpose of statistical analysis. Depicting all measurements is useful, but statistical comparisons should use medians or means.

5. Minor point (grammar): even though RAD51 foci are multiple, the singular should be used in the expression of 'RAD51 focus formation' - but indeed plural in 'the formation of RAD51 foci' in analogy with 'sample collection' vs 'collection of samples'.

Reviewer #3 (Remarks to the Author):

In the revised version of the manuscript, Salas-Lloret et al. have addressed some of the issues raised in the original version of the work. The results strongly support the identification of PCNA K164 as a target for the E3 ubiquitin ligase activity of the BRCA1-BARD1 complex. In addition, the authors show that the catalytically inactive BRCA1 I26A mutant leads to the accumulation of ssDNA gaps, and is associated with resistance to PARP inhibitors and sensitivity to HU. In contrast, the relevance of this activity in vivo is still not clearly established: the contribution of BRCA1-BARD1 to the basal levels of PCNA ubiquitylation remains to be determined and the direct role of BRCA1-BARD1 in the generation of ssDNA gaps during DNA replication through the control of PCNA ubiquitylation is not convincingly proven.

PCNA is identified as a substrate of BARD1 that requires the presence of BRCA1 by TULIP assay. Consistently, the loss of BRCA1 leads to reduced levels of PCNA K164Ub and this ubiquitylation is partially rescued with WT BRCA1. Unexpectedly, a similar rescue is observed with the catalytically inactive BRCA1 I26A mutant, suggesting that this mutant might support the ubiquitylation of PCNA by BARD1 even if it cannot modify histone H2A. The authors claim that the rescue is higher with the WT BRCA1 than with I26A is not supported by the quantification of individual experiments shown in Figure S3D-E. These differences could be driven by changes in the cell cycle (not analyzed by the authors although the experiment was suggested in the first revision) or by basal DNA damage/replication fork collapse (suggested by the authors, not assayed). Still, the issue remains and it is not clear if the I26A mutant supports PCNA ubiquitylation.

Regarding the contribution of the different E3 ligases, the authors show that RAD18 can take over in the absence of BRCA1, showing that both ligases are relevant. However, the experiments included in the manuscript and in the response to other reviewers do not clarify the relative contribution of these E3 ligases to PCNA K164Ub in unperturbed conditions.

Most of the functional experiments rely on the use of the I26A mutant which, as explained before, shows a similar rescue of PCNA ubiquitylation as WT BRCA1. The authors clearly demonstrate that the I26A mutant confers resistance to PARP inhibitors but it cannot rescue the sensitivity to HU. There are some important inconsistencies in the interpretation of the results:

1. The I26A mutant cannot rescue the accumulation of ssDNA gaps induced by the lack of BRCA1, similar to PCNA K164R. Since the levels of ubiquitylated PCNA are similar in WT and I26A complemented cells, it is not possible to conclude that the effect is mediated by PCNA ubiquitylation.
2. BRCA1 KO cells are sensitive to HU treatment and show increased ubiquitylation of PCNA in response to the treatment. Cells complemented with WT BRCA1 are resistant to HU and show lower levels of PCNA ubiquitylation than BRCA1 KO cells. Again, changes in PCNA ubiquitylation would not justify the sensitivity/resistance to HU. As the authors discuss, the difference in sensitivity could be due to DSB generated upon HU treatment.
3. The I26A mutant phenocopies the increase in ssDNA in BRCA1 KO cells in unperturbed conditions (Figure 2C). In the presence of HU, in contrast, the increase in ssDNA in BRCA1 KO cells is much greater than in I26A complemented cells (Figure S2B). Again, the changes in PCNA ubiquitylation do not follow the levels of ssDNA gaps.

Therefore, it is clear that PCNA is ubiquitylated by the BRCA1-BARD1 complex. In addition, the authors show that the I26A mutant leads to the accumulation of ssDNA gaps. However, the data do not justify the strong claims of the authors for a link between PCNA ubiquitylation by the BRCA1-BARD1 complex, the induction of ssDNA gaps and the sensitivity to HU. In this same line, the authors provide an experiment in page 7 of the rebuttal where neither depletion of BRCA1 nor rescue with WT or I26A changes the sensitivity to HU. These experiments are discarded as inconclusive with no real basis. In my opinion, these experiments are much stronger evidence showing that the action of

BRCA1-BARD1 to control HU sensitivity is mediated by PCNA ubiquitylation. If this is the case, changing BRCA1 in a K164R background should have no effect, as shown in the figure. However, there are still many possible interpretations of the results. The conclusions should be toned down and the other options considered.

The quantification of ubiquitylated PCNA is central to the manuscript and it is still not very convincing. The loading is uneven (not for the Ponceau though), the membranes are cut very close to the bands or even through the bands, and the data show a very high dispersion. In my opinion Figure S3 should be a main figure and is still not solid enough. There is no reason to need cutting the membrane to measure PCNA and mono-Ub-PCNA. Avoiding this step should help getting more consistent results.

Similarly, the authors still show fiber assays that have been only repeated twice. This could be justifiable for some accessory experiments but again, these assays are central for the functional link in the manuscript. The statistical analysis of the S1 nuclease assays are OK for the individual experiments but it only represents the consistency within one experiment. Still the representation of the mean of the medians and the statistical analysis using the t-test or ANOVA is necessary, using at least 3 independent experiments.

Other additional comments:

1. Figure S1E, TULIP should be indicated to show that the overexpressed protein is the fusion.
2. Figure 2B, the model is misleading as it shows PCNA ubiquitylation with the I26A mutant.
3. Figure S2A is not called in the text.
4. Line 200 "BrU" instead of "BrdU"
5. Line 343 "and" instead of "an"

We thank the reviewers for the time employed in reviewing the manuscript and for the feedback provided. This feedback has prompted us to re visit and re-interpretate our data, resulting in significant improvements to the manuscript.

Overall, we would like to emphasize that the main message of the manuscript is about what happens in unperturbed conditions. The data related to the response to Hydroxyurea is accessory and its inclusion is justified because the experiments have been performed and that data might be of use for the community.

REVIEWER COMMENTS

Reviewer #1 (Remarks to the Author):

The revised manuscript addressed my comments in a satisfactory manner. In particular, the authors investigated the formation of ssDNA gaps under unperturbed conditions in RAD18-knockout cells, performed epistasis experiments, and clarified the ubiquitination of endogenous PCNA. In addition, they clarified a number of issues I had raised. Overall, in my opinion, the manuscript is suitable for publication in Nature Communications.

We thank this reviewer for the support for publication. Nevertheless, we have included most of the experiments performed for the previous rebuttal in the manuscript as suggested by the other two reviewers and toned down our claims about the endogenous PCNA ubiquitination.

Reviewer #2 (Remarks to the Author):

The authors have addressed most of the referees' comments and made significant improvements to the manuscript. My general opinion is that these improvements do not quite go far enough, and some of the new data and arguments, even if contradictory, should be included in the manuscript.

We have included most of the new data and arguments in the manuscript. Note that the manuscript was submitted as a short article and the intended message of the manuscript is what is stated in the title "BRCA1/BARD1 ubiquitinates PCNA in unperturbed conditions to promote continuous DNA synthesis".

Specific points:

1. I welcome the addition to the introduction, but suggest the following modification:
“...as _THE EFFECTS OF_ patient-derived BRCA1 RING domain mutations such as C61G or C64G which are required for tumor suppression and HR are likely due to the disruption of the interaction of BRCA1 with BARD1”

We have modified this sentence as suggested by the reviewer.

2. Regarding the new quantified data in figure S3, it is surprising that PCNA ubiquitylation is apparently induced by HU in BRCA1 mutant cells, as much or even more than in the rescued ones. (Fig. S3C), and more than in those expressing the I26A mutant. The remark that “yet these levels were lower than in the other three cell lines” (which is not supported by statistics, and should reference Fig. S3E) provides little explanation for this result that does not fit with the model. It is also a problem that no statistics are provided in any part of Fig. S3. This set of experiments should be

performed in a sufficient number of variations (e.g. including other drugs) to provide a convincing dataset for main figures with significant differences showing that PCNA ubiquitylation in some specific settings is dependent on BRCA1 ubiquitin ligase activity - otherwise it must be made clear that this has not been proven.

We have become now aware by these comments (combined with the comments from Reviewer 3) that we failed in communicating the question that we wanted to answer with these experiments.

In this manuscript we wanted to address the function of BRCA1/BARD1 E3 activity in unperturbed conditions, thus, the HU experiments are accessory. We found it intriguing that the I26A mutant was resistant to Olaparib-induced gaps but not to HU, for that reason, we look at the PCNA ubiquitination levels in response to these drugs. Therefore, we believe that assaying PCNA-Ub levels to a variety of drugs is out of the scope of the manuscript.

The relevant analysis is the levels of PCNA-Ub levels in unperturbed conditions (Now Supplementary Figure 3E) where we see a decrease of PCNA-Ub levels in BRCA1-KO compared to Parental Cells which is partially complemented by the BRCA1-WT constructs. Interestingly, BRCA1-I26A can complement at the same level than the WT construct. This indicates: 1) that the fraction of ubiquitinated PCNA relevant to suppress ssDNA formation is negligible from the total PCNA-Ub levels, yet highly functionally relevant and 2) that the decrease observed in BRCA1-KO is mainly due to replication fork collapses.

Additionally, the decrease in the PCNA-Ub fraction in RAD18-KO cells might also be due to transcription-associated replication fork catastrophe and yet the remaining PCNA-Ub fraction is sufficient to prevent ssDNA gap formation.

Thus, we believe that is not possible to estimate which percentage of PCNA-Ub in unperturbed conditions corresponds to which E3 by performing KOs, as compensatory mechanisms can arise.

All these arguments are now discussed in the manuscript.

3. The authors provide experimental data in the rebuttal showing a 75% reduction of PCNA-Ub in unperturbed cells upon knockout of RAD18. I presume these are RPE-1 p53^{-/-} cells, but these data do not seem to be in the manuscript. At the same time, they could not find RAD18-TULIP-PCNA conjugates in unperturbed cells, only upon UV treatment. This is a clear discrepancy, which could be scientifically interesting. The data provided in the revision should be included in the paper and discussed. In the revision the authors argue that the KO of p53 could interfere with the results, but I do not really see the argument. Aren't all tested cell lines RPE-1 p53^{-/-}? The 'parental' cell line should be precisely specified in the methods and in the figure legends.

WE have now discussed these apparent discrepancies. In brief, RAD18-KO produces transcription-dependent fork collapse and BRCA1-KO also produces fork collapse upon stress. Thus, the reduction in BRCA1-KO and RAD18-KO can be due to fork collapse upon encountering obstacles which would be overcome with the participation of these E3s. However, our TULIP2 assays enable us to overcome these hurdles as they are performed in conditions where all the E3s are present and at expression levels below endogenous, avoiding the appearance of overexpression artefacts. Therefore, we believe that our TULIP2 assays are more informative of what is happening constitutively in the cells.

Indeed, all the tested cell lines are RPE1 TP53^{-/-}. In the previous rebuttal we indicated that BRCA1 and RAD18 are essential genes and, for that reason, the additional TP53-KO mutation was required to generate BRCA1-KO and RAD18-KO cells, respectively. We did not mean that the KO of P53 could interfere with the results. We have clarified in the methods section of the manuscript that all the Parental RPE1 cells used in the manuscript are TP53^{-/-}.

4. The new epistasis experiment in Fig. 3A is useful, as is the inhibition of pol theta in Fig. 3B. However, I must echo reviewer 3 in drawing attention to the bad practice that fibres from the same experiment are treated as individual data points for the purpose of statistical analysis. Depicting all measurements is useful, but statistical comparisons should use medians or means.

As explained in the previous rebuttal, we prefer to stick to the analysis methodology that is used in the previous literature and indicated in the Methods paper from the Vindigni lab. As an intermediate approach, we have indicated the medians of each independent experiment in the figures. Moreover, at the end of this rebuttal, which will be published alongside with the paper, the statistical analysis using the medians of each independent experiments as replicates are attached (N=3; RM ANOVA analysis).

As expected, the p-values are worse than in the previous analysis, but the conclusions remain. Given that these are very costly experiments, we believe that performing additional replicates to improve these p-values with this statistical methodology does not add up to the manuscript and will not improve nor change the message.

5. Minor point (grammar): even though RAD51 foci are multiple, the singular should be used in the expression of 'RAD51 focus formation' - but indeed plural in 'the formation of RAD51 foci' in analogy with 'sample collection' vs 'collection of samples'.

We have corrected this grammar point as indicated by the reviewer.

Reviewer #3 (Remarks to the Author):

In the revised version of the manuscript, Salas-Lloret et al. have addressed some of the issues raised in the original version of the work. The results strongly support the identification of PCNA K164 as a target for the E3 ubiquitin ligase activity of the BRCA1-BARD1 complex. In addition, the authors show that the catalytically inactive BRCA1 I26A mutant leads to the accumulation of ssDNA gaps, and is associated with resistance to PARP inhibitors and sensitivity to HU. In contrast, the relevance of this activity in vivo is still not clearly established: the contribution of BRCA1-BARD1 to the basal levels of PCNA ubiquitylation remains to be determined and the direct role of BRCA1-BARD1 in the generation of ssDNA gaps during DNA replication through the control of PCNA ubiquitylation is not convincingly proven.

We thank this reviewer for these remarks. This reviewer agrees that our results support the main claims of the manuscript. The main concern of the reviewer are the results related to the response to HU, which are accessory experiments as our manuscript aims to obtain insight of what happens in unchallenged conditions.

Thus, we respectfully disagree with this reviewer about the relevance for the message of the manuscript of determining which percentage of the PCNA ubiquitinated fraction corresponds to which E3. We agree with the reviewer that our data strongly support the

identification of PCNA as a substrate of BRCA1/BARD1 and lack of this E3 activity promotes the accumulation of ssDNA gaps.

Nevertheless, as the main concern of this reviewer is the contribution of each E3 enzyme to the basal levels of PCNA ubiquitination, aiming to provide more insight, we have revisited our data and the literature. We have to conclude that this cannot be addressed by performing knockouts and western blots as PCNA ubiquitination occurs in different scenarios, by different E3 enzymes which are likely to occur simultaneously in the same cell culture or even in the same cell. We have toned down our conclusions regarding this aspect and provided more discussion in the new version of the manuscript.

PCNA is identified as a substrate of BARD1 that requires the presence of BRCA1 by TULIP assay. Consistently, the loss of BRCA1 leads to reduced levels of PCNA K164Ub and this ubiquitylation is partially rescued with WT BRCA1. Unexpectedly, a similar rescue is observed with the catalytically inactive BRCA1 I26A mutant, suggesting that this mutant might support the ubiquitylation of PCNA by BARD1 even if it cannot modify histone H2A. The authors claim that the rescue is higher with the WT BRCA1 than with I26A is not supported by the quantification of individual experiments shown in Figure S3D-E. – We have done statistics in this regard and agree with this reviewer. We have removed this claim. We now think that the contribution of BRCA1/BARD1 to the basal levels of PCNA are negligible from the total levels, yet highly relevant to suppress ssDNA formation, accordingly, the very low levels of PCNA ubiquitination remaining after RAD18-KO suffice to promote ssDNA gap suppression.

These differences could be driven by changes in the cell cycle (not analyzed by the authors although the experiment was suggested in the first revision) – The cell cycle of the different cell lines have been analyzed and indeed, differences in the level of unmodified PCNA can be explained by differences in the cell cycle (PCNA expression occurs in S-phase). In fact, the lower proportion of cells in S-phase in the Parental cells can explain the apparent uneven loading observed by lower levels of unmodified PCNA but not in the Ponceau (as mentioned below by this reviewer). However, the levels of PCNA ubiquitination are normalized to the levels of unmodified PCNA, thus correcting for this aspect. or by basal DNA damage/replication fork collapse (suggested by the authors, not assayed) – The role of BRCA1 preventing replication fork collapse have been previously described by multiple authors. Moreover, our IPOND experiments show accumulation of MDC1 at the forks in BRCA1-KO cells which is indicative of damage at the forks. Still, the issue remains, and it is not clear if the I26A mutant supports PCNA ubiquitylation. – In the I26A, there is no response to HU in terms of PCNA ubiquitination (Supplementary Figure 3D) while RAD51 loading is not affected (Figure 1B,C).

Regarding the contribution of the different E3 ligases, the authors show that RAD18 can take over in the absence of BRCA1, showing that both ligases are relevant. However, the experiments included in the manuscript and in the response to other reviewers do not clarify the relative contribution of these E3 ligases to PCNA K164Ub in unperturbed conditions.

We showed, that in the absence of BRCA1, the activity of RAD18 towards PCNA is increased by several orders of magnitude. However, although RAD18 taking over BRCA1 would be a possibility, not being able to rescue gap-suppression suggest as a more likely scenario that the lack of BRCA1 results in the accumulation of structures that require the activity of RAD18 towards PCNA.

Most of the functional experiments rely on the use of the I26A mutant which, as explained before, shows a similar rescue of PCNA ubiquitylation as WT BRCA1. The authors clearly demonstrate that the I26A mutant confers resistance to PARP inhibitors but it cannot rescue the sensitivity to HU. There are some important inconsistencies in the interpretation of the results:

1. The I26A mutant cannot rescue the accumulation of ssDNA gaps induced by the lack of BRCA1, similar to PCNA K164R. Since the levels of ubiquitylated PCNA are similar in WT and I26A complemented cells, it is not possible to conclude that the effect is mediated by PCNA ubiquitylation.

We have now discussed this. We think that, since the total levels of ubiquitinated PCNA are similar in the BRCA1-WT and BRCA1-I26A rescued cells, the fraction of ubiquitinated PCNA that prevents the formation of ssDNA gaps is negligible from the total amount of Ubiquitinated PCNA. PCNA can be ubiquitinated by different E3s in different contexts. In 2021, the Vindigni lab showed that UBE2N promote gap filling in S and Rad18 in G2 (Tirman *et al*, 2021). However, they did not link this gap filling (or suppression) phenotype of BRCA1 to PCNA ubiquitination nor BRCA1 E3 activity.

2. BRCA1 KO cells are sensitive to HU treatment and show increased ubiquitylation of PCNA in response to the treatment. Cells complemented with WT BRCA1 are resistant to HU and show lower levels of PCNA ubiquitylation than BRCA1 KO cells. Again, changes in PCNA ubiquitylation would not justify the sensitivity/resistance to HU. As the authors discuss, the difference in sensitivity could be due to DSB generated upon HU treatment.

Indeed, we have now discussed this. We think that the sensitivity of the BRCA1-KO is a due to the occurrence of DSBs as previously mentioned and illustrated in Figure 5. In a BRCA1-KO the filling of gaps can be performed in alternative pathways by other E3s modifying PCNA (including RAD18 but not excluding other E3s). In the case, of PCNA-K164R, the sensitivity arises from no E3 being able to modify it, including BRCA1, RAD18 and other yet to described E3s, but DSBs can still be repaired by HR.

Consistently, the I26A can repair the DSBs, and other E3s can modify PCNA in alternative pathways to fill the gaps. Thus, the sensitivity is milder than both PCNA-K164R and BRCA1-KO.

3. The I26A mutant phenocopies the increase in ssDNA in BRCA1 KO cells in unperturbed conditions (Figure 2C). In the presence of HU, in contrast, the increase in ssDNA in BRCA1 KO cells is much greater than in I26A complemented cells (Figure S2B). Again, the changes in PCNA ubiquitylation do not follow the levels of ssDNA gaps.

In the BRCA1-KO cells, replication is less restrained than in the in the I26A mutant (I26A fork restraining resembles more the levels PCNA-K164R). The longer the fork speed, the longer the DNA fiber and higher the probability of gaps happening.

Therefore, it is clear that PCNA is ubiquitylated by the BRCA1-BARD1 complex. In addition, the authors show that the I26A mutant leads to the accumulation of ssDNA gaps. However, the data do not justify the strong claims of the authors for a link between PCNA ubiquitylation by the BRCA1-BARD1 complex, the induction of ssDNA gaps and the sensitivity to HU. In this same line, the authors provide an experiment in page 7 of the rebuttal where neither depletion of BRCA1 nor rescue with WT or I26A changes the sensitivity to HU. These experiments are discarded as inconclusive with no real basis. In my opinion, these experiments are much stronger evidence showing that the action of BRCA1-BARD1 to control HU sensitivity is mediated by PCNA ubiquitylation. If this is the case, changing BRCA1 in a K164R background should have no effect, as shown in the figure. However, there are still many possible interpretations of the results. The conclusions should be toned down and the other options considered.

We thank this reviewer for pointing this out, we have included the experiments in the manuscript (new Figures 3D-E).

As afore mentioned, we have toned down our conclusions and discussed apparent discrepancies in the data in the manuscript.

The quantification of ubiquitylated PCNA is central to the manuscript and it is still not very convincing. The loading is uneven (not for the Ponceau though), the membranes are cut very close to the bands or even through the bands, and the data show a very high dispersion. In my opinion Figure S3 should be a main figure and is still not solid enough. There is no reason to need cutting the membrane to measure PCNA and mono-Ub-PCNA. Avoiding this step should help getting more consistent results.

We insist that the relative contribution of each E3 enzyme to the basal levels of ubiquitinated PCNA is not the message of the manuscript. Knocking out E3s might have collateral effects which may affect PCNA ubiquitination levels, for example, we think that the decrease observed in BRCA1-KO cells is mainly due to fork collapse. In the case of RAD18, RAD18 also travels with the fork (IPOND data) and it has been shown that is required to overcome genome instability associated to transcription-replication conflicts and its absence causes transcription-dependent fork collapse (Wells *et al*, 2022). Thus, there is the possibility that the decrease in PCNA-Ub observed in RAD18-KO is also, at least partially, due to fork collapse.

Given the high number of E3s that have been described to modify PCNA, plus the high number of E3s in the human proteome that remain uncharacterized, we think that it is not possible to draw conclusions from measuring endogenous PCNA ubiquitination levels after knockout of different E3s. Among these uncharacterized E3s, recently, in a search for substrates for the poorly characterized E3 RNF214, PCNA also appeared as a substrate for RNF214 in a high throughout screen (Barroso-Gomila *et al*, 2023), but deficiencies in RNF214, neither RAD18, does not cause sensitivity to HU (Olivieri *et al*, 2020).

Importantly, our TULIP methodology enables us to measure E3 activity towards targets in a scenario where all the other E3s are present, thus overcoming the problem of collateral effects of absence of essential genes, and at levels below the endogenous, thus overcoming the problem of appearance of overexpression artefacts.

All these arguments are now included in the discussion.

Regarding, the reason to cut the membrane, we have explained that this is necessary to avoid signal interference, to enable quantification, to avoid the non-modified band to

sequester most of the antibody and for the correct visualization of the modified band in overexposed blots. We do not cut the membranes too close to the bands, and absolutely not across the bands. We are not the first ones to face the same problem and adopt the same solution. In a previous collaborative paper by the labs of Massimo Lopes, Lorenza Penengo, Alberto Ciccia and David Cortez the same solution was adopted to overcome the same problem (Vujanovic *et al*, 2017).

Nevertheless, we attach here an example of what occurs if we do not cut the membrane. For this, we have re-run again the samples from one of the repeats in Figure 3D.

Similarly, the authors still show fiber assays that have been only repeated twice. This could be justifiable for some accessory experiments but again, these assays are central for the functional link in the manuscript. The statistical analysis of the S1 nuclease assays are OK for the individual experiments but it only represents the consistency within one experiment. Still the representation of the mean of the medians and the statistical analysis using the t-test or ANOVA is necessary, using at least 3 independent experiments.

As previously explained to reviewer 2, who agrees with this reviewer in this aspect, and as explained in the previous rebuttal, we prefer to stick to the analysis methodology that is used in the previous literature and indicated in the Methods paper from the Vindigni lab. As an intermediate approach we have indicated the medians of each independent experiment in the figures. Moreover, at the end of this rebuttal, which will be published alongside with the paper, the statistical analysis using the medians of each independent experiments as replicate are attached (N=3; RM ANOVA analysis).

Individually, within each experiment, the differences were highly significant as well.

As expected, the p-values are worse than in the previous analysis, but the conclusions remain the same. Given that these are very costly experiments, we believe that performing additional replicates to improve these p-values with this statistical methodology does not add up to the manuscript and will not improve nor change the message.

Other additional comments:

1. Figure S1E, TULIP should be indicated to show that the overexpressed protein is the fusion.

We have modified this accordingly. Additionally, we have also indicated it in S1F and S1G

2. Figure 2B, the model is misleading as it shows PCNA ubiquitylation with the I26A mutant.

We appreciate the feedback provided by this reviewer and modified it accordingly.

3. Figure S2A is not called in the text.

We have corrected this.

4. Line 200 “BrU” instead of “BrdU”

Thanks for pointing out this typo. We have corrected it accordingly.

5. Line 343 “and” instead of “an”

Thanks for pointing out this typo. We have corrected it accordingly.

References

Barroso-Gomila O, Merino-Cacho L, Muratore V, Perez C, Taibi V, Maspero E, Azkargorta M, Iloro I, Trulsson F, Vertegaal ACO *et al* (2023) BioE3 identifies specific substrates of ubiquitin E3 ligases. *Nature communications* 14: 7656

Olivieri M, Cho T, Alvarez-Quilon A, Li K, Schellenberg MJ, Zimmermann M, Hustedt N, Rossi SE, Adam S, Melo H *et al* (2020) A Genetic Map of the Response to DNA Damage in Human Cells. *Cell* 182: 481-496 e421

Tirman S, Quinet A, Wood M, Meroni A, Cybulla E, Jackson J, Pegoraro S, Simoneau A, Zou L, Vindigni A (2021) Temporally distinct post-replicative repair mechanisms fill PRIMPOL-dependent ssDNA gaps in human cells. *Molecular cell* 81: 4026-4040 e4028

Vujanovic M, Krietsch J, Raso MC, Terraneo N, Zellweger R, Schmid JA, Tagliatala A, Huang JW, Holland CL, Zwicky K *et al* (2017) Replication Fork Slowing and Reversal upon DNA Damage Require PCNA Polyubiquitination and ZRANB3 DNA Translocase Activity. *Molecular cell* 67: 882-890 e885

Wells JP, Chang EY, Dinatto L, White J, Ryall S, Stirling PC (2022) RAD18 opposes transcription-associated genome instability through FANCD2 recruitment. *PLoS Genet* 18: e1010309

REVIEWERS' COMMENTS

Reviewer #2 (Remarks to the Author):

The manuscript has been substantially improved following the second revision with additional calculations and clarified discussions of the data, and on the whole it is now suitable for publication. The results will be of substantial interest to the DNA replication and repair field.

The data in Fig. S3E on the levels of PCNA-Ub in unperturbed control and BRCA1 mutant cells is now correctly described in the results section. I believe that the discussion of these data still contains over-interpretation, i.e. that the lost function of BRCA1 in fork stability is causing the reduction in PCNA-Ub - 'indicates' should be changed to 'suggests' in line 363 to allow for other possibilities.

Finally, I would like to request the authors to include the new statistical analysis of the DNA fibre assay in the manuscript. The authors write that "the p-values are worse than in the previous analysis" - but they do concur with the observed effects, and they are biologically meaningful, unlike the treatment of each fibre as an independent measurement.

Reviewer #3 (Remarks to the Author):

The revised version of the manuscript by Salas-Lloret et al. has incorporated new experiments that were previously included in the response to the reviewers. The conclusions have been toned down (as the effect of HU) and alternative explanations to some of the results have been discussed. As indicated in my previous review, the authors show that PCNA is a substrate for the BRCA1/BARD1 heterodimer and that the function is relevant for the basal levels of PCNA ubiquitylation. I think that the manuscript is now suitable for publication although I strongly suggest including additional changes to take into account possible alternative explanations for the data and to provide a better explanation of the proposed model.

At this point the manuscript includes different contradictory statements. In lines 272-276 the authors propose that "the contribution of BRCA1 E3 activity to the total levels of ubiquitinated PCNA is neglectable". Then, in lines 305-307 and 334-335 the authors claim that BRCA1-BARD1 is the main E3 ligase for PCNA in unperturbed conditions. This is justified by a secondary effect of BRCA1 KO and I26A in the induction of fork collapse that indirectly reduces PCNA ubiquitylation. To reconcile these results, the authors propose that BRCA1/BARD1 is not the main E3 ligase for PCNA in unperturbed conditions but instead targets a small fraction of PCNA relevant for ssDNA gap generation. This should be made clear through the text.

Other hypotheses can explain the control of basal PCNA ubiquitylation and its role in ssDNA gap generation during DNA replication. In fact, the data of BRCA1 and RAD18 KO cells show that they both lead to decreases in PCNA-ubiquitylation higher than 50%. This could indicate that there is cross-talk between both processes, for instance. These alternative explanations should be taken into account.

I recommend to clarify/tone down:

1. Line 88, I would tone down the sentence "to solve the controversy in the field".
2. Lines 206-207, a partial rescue of ssDNA gaps by BRCA1-I26A-GFP can be seen in Figure 2C, this statement should be toned down.
3. Lines 222-223, similar to point 2 there is a partial rescue of ssDNA by BRCA1-I26A in Figure S2B. The statement should be toned down.
4. Lines 253-259. I do not see an additional effect of shBRCA1 in K164R cells. The explanation of the effect of sh resistant WT and I26A BRCA1 is very confusing and needs to be rephrased.
5. Supplementary Figure 4A shows a line in ubPCNA and PCNA that is touching the bands and

corresponds to the cut in the membrane. As indicated in the previous revision, the quality of this blot is not good enough.

6. Lines 305-307 and 334-335 should be rephrased to better explain the hypothesis behind this interpretation of the data.

7. Lines 376-378, this explanation is not clear. The authors should just explain how the KO of specific E3 ligases might influence the basal levels of ubiquitinated PCNA.

Typos:

1. Line 370, fork instead of for.
2. Line 371, explain instead of explains.